# Multi-Objective Preference Optimization:
# Improving Human Alignment of Generative Models

**Akhil Agnihotri** [1]  **Rahul Jain** [1 2]  **Deepak Ramachandran** [2]  **Zheng Wen** [2 3]

## Abstract

Post-training LLMs with RLHF and preference optimization methods (e.g., DPO, IPO) has greatly improved alignment, yet these approaches assume a single objective. In reality, humans express multiple, often conflicting objectives, such as helpfulness and harmlessness, with no natural scalarization. We study the multi-objective preference alignment problem, where a policy must balance several objectives simultaneously. We propose Multi-Objective Preference Optimization (`MOPO`), a constrained KL-regularized framework that maximizes a primary objective while enforcing lower bounds on secondary objectives via tunable safety thresholds. `MOPO` operates directly on pairwise preferences without point-wise rewards, and admits simple closed-form iterative updates. Empirically, `MOPO` recovers Pareto-optimal policies on synthetic benchmarks and, when fine-tuned on human-preference data, yields multi-billion parameter models that achieve higher rewards and Pareto-dominate baselines, with stable and robust optimization dynamics.

## 1. Introduction

Aligning Large Language Models (LLMs) and other generative models with human preferences (Ouyang et al., 2022; Rafailov et al., 2023; Gheshlaghi Azar et al., 2024) has evolved from single-objective to multi-objective (Rame et al., 2023; Yang et al., 2024; Zhong et al., 2024), aiming to comprehensively capture the inherent heterogeneity of human preferences. Multi-objective alignment jointly considers multiple human preference objectives, such as safety, helpfulness, factuality, and diversity, to optimize the LLM. However, human preferences are neither one di-

mensional nor fixed: a single conversation may demand an answer that is simultaneously helpful, harmless, concise, and imaginative. Optimizing an LLM for just one of these axes often degrades the others, as seen between helpfulness and safety tasks (Bai et al., 2022; Rame et al., 2023; Yang et al., 2024; Chitlangia et al., 2021). Consequently, we seek multi-objective alignment, where the model is judged by reward vector performance on possibly conflicting objectives, rather than a single scalarized score.

From an optimization standpoint this setting is naturally cast as multi-objective optimization (Miettinen, 1999; Keeney & Raiffa, 1993), where desirable solutions form a Pareto front. Existing alignment pipelines largely collapse this vector into a weighted sum and run reinforcement learning from human feedback (RLHF) on the resulting scalar reward (Ouyang et al., 2022; Rafailov et al., 2023), but linear or non-linear scalarization hides important corner cases, and requires re-training for each weight choice. To overcome this, there is some work on decoding at inference time, but it still aims to cast user preferences as vector inputs to the model (Shi et al., 2024; Wang et al., 2024c; Hu et al., 2022b). However, users seldom articulate explicit weights; they expect models to adapt at low latency. These observations motivate algorithms that (i) approximate the Pareto front *offline*, and (ii) provide inference without probing users for preference vector inputs. Motivated by these insights, we introduce `MOPO`, an offline constrained-optimization framework that delivers multi-objective alignment with a single multi-head policy.

We begin by formulating the problem as a concave constrained optimization problem where preferences along the 'primary' objective are maximized while preferences along the 'secondary' objectives are constrained above a tunable threshold. We then motivate bounding the *lower bound* of preferences (instead of the naively constraining these secondary objectives), followed by a behaviour cloning approach to extract the optimal policy from the resulting optimal importance sampling ratio. Overall, this procedure results in iterative updates of the underlying optimization variables, which is scalable and robust to the hyperparameters. Our contributions are as follows: (i) We propose `MOPO`, an offline constrained optimization based preference-only

[1]University of Southern California [2]Google DeepMind [3]OpenAI. Correspondence to: Akhil Agnihotri <agnihotri.akhil@gmail.com>.

*Proceedings of the $43^{rd}$ International Conference on Machine Learning*, Seoul, South Korea. PMLR 306, 2026. Copyright 2026 by the author(s).

learning algorithm that optimizes for multiple objectives and achieves Pareto optimality. (ii) We empirically validate the correctness on a variety of canonical preference dataset types that show how `MOPO` approximates the Pareto front when it is known. (iii) We conduct extensive LLM experiments on 7B parameter models with real world data to validate the effectiveness of `MOPO`, and show optimization stability through ablations.

**Related works.** `RLHF` (Christiano et al., 2017; Ziegler et al., 2019) has become the de-facto paradigm for aligning LLMs such as GPT-4 (Achiam et al., 2023) and LLaMA-3 (Grattafiori et al., 2024). Most `RLHF` pipelines fit a reward model to pairwise preferences and then fine-tune the policy with PPO (Schulman et al., 2017; Ouyang et al., 2022). Instability and sample inefficiency have motivated alternatives that still target a scalar reference-regularized objective, including RAFT (Zhang et al., 2024), RRHF (Yuan et al., 2023), DPO (Rafailov et al., 2023), ΨPO (Gheshlaghi Azar et al., 2024), and Nash-`RLHF` (Munos et al., 2023). These methods (except ΨPO) inherit a fundamental limitation: all preferences are collapsed into a single reward signal, obscuring trade-offs between objective(s). Recent works attempt to optimize multiple objectives by learning scalarization functions or prompt contexts (Hu et al., 2023; Zhong et al., 2024; Agnihotri et al., 2019; Guo et al., 2024; Wu et al., 2023; Li et al., 2025; Lee et al., 2025). Although effective in specific domains, such approaches seldom achieve Pareto-optimal solutions even when the Pareto front is known (Yang et al., 2024; Mukherjee et al., 2024). While (Rame et al., 2023) mitigate tuning via 'Rewarded Soups', and MORLHF (Li et al., 2020) and MODPO (Zhou et al., 2023) borrow ideas from multi-objective RL, they still learn with respect to a single functional combination of rewards. RiC (Yang et al., 2024), HaM (Mukherjee et al., 2024), MOD (Shi et al., 2024), and DPA (Wang et al., 2024c) move beyond heuristic scalarization by conditioning on multiple rewards at inference time. While these algorithms improve controllability, they still rely on inference-time user preference input to optimize multiple objectives, which can misrepresent complex preference structures and are hard to quantify practically (for instance, what does "0.6 helpful, 0.4 safe" imply?). PARM (Lin et al., 2025) trains a *single*, preference conditioned policy across multiple objectives, however, it still requires user preference inputs. A contemporary work to ours is L3Ms (Dhillon et al., 2025), which uses log barrier functions to incorporate constraints into the primary alignment problem. Classical multi-objective RL (MORL) focuses on discovering Pareto-efficient policies under vector rewards (Roijers et al., 2013; Van Moffaert & Nowé, 2014; Hayes et al., 2022). Constrained MORL (Huang et al., 2022; Liu et al., 2024; Agnihotri et al., 2024a;b) methods use scalar rewards to maximize a primary objective while enforcing lower bounds on secondary objectives.

## 2. Preliminaries

We first begin with a motivating example that inspires development of `MOPO` as an offline constrained optimization algorithm learning from preference data. Following this, we describe the problem setting with a formal problem statement. Throughout the main text, we keep notation light and refer the reader to Appendix A for a complete discussion, where we prove that current literature as discussed above fails to achieve Pareto-optimality.

**A Motivating Example.** In this example, we empirically demonstrate the necessity of *principled* multi-objective optimization methods that account for multi-dimensional preferences. We benchmark various approaches for multi-objective alignment and show that existing state-of-the-art techniques consistently fail to reach the Pareto front. To ensure clarity, we conduct experiments on synthetic datasets where true Pareto front is known, allowing for precise evaluation of alignment quality. Due to space constraints we keep discussion concise, and refer the reader to Appendix A.1 for completeness.

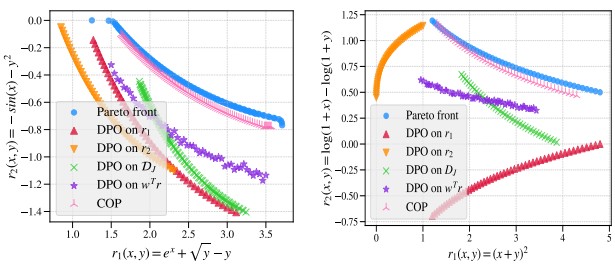

*(a)* `COP` comparison with traditional approaches

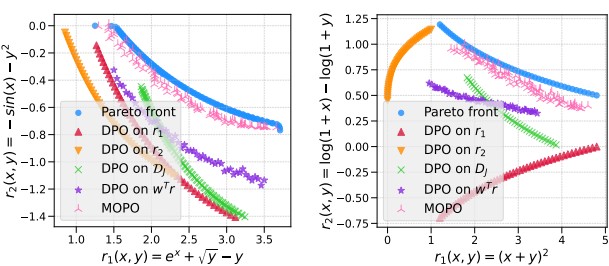

*(b)* `MOPO` approximates the Pareto front

*Figure 1.* Illustration of how a `COP` approach, and hence `MOPO`, achieves Pareto-optimal alignment.

Consider this toy preference example with input and action spaces $\mathcal{X} = \mathcal{Y} = [0,1]$. For any triplet $(x, y, y')$ we draw a preference label $z \in \{y, y'\}$ from the Bradley-Terry model $BT(r)$ with $\Pr[z = y \mid x, y, y'] = \exp(r(x,y))/(\exp(r(x,y)) + \exp(r(x,y')))$, where $r(\cdot)$ is the underlying reward model (Bradley & Terry, 1952). We study two bi-objective settings: *Set A* with $r_1 = e^x + \sqrt{y} - y$ and $r_2 = -\sin x - y^2$, and *Set B* with $r_1 = (x + y)^2$ and $r_2 = \log((1 + x)/(1 + y))$. From i.i.d. samples $(x, y, y') \sim \mathcal{U}([0,1]^3)$ we construct four datasets: (i) $\mathcal{D}_1$,

labeled by $r_1$ only; (ii) $\mathcal{D}_2$, by $r_2$ only; (iii) $\mathcal{D}_J$, the **j**oint dataset retaining samples where the two labels coincide; and (iv) $\mathcal{D}_C$, the **c**ombined dataset labeled by $\mathrm{BT}(wr_1 + (1-w)r_2)$ for $w \in [0,1]$. As seen in Figure 1(a), policies trained with DPO on $\mathcal{D}_1$ or $\mathcal{D}_2$ alone ignore one objective, those trained on $\mathcal{D}_J$ see only non-conflicting pairs, and those trained on $\mathcal{D}_C$ are biased toward a single scalarization of reward functions, so all three miss large portions of the Pareto front. A constrained optimization baseline $\pi_{\mathrm{COP}}(x) = \mathrm{argmax}_y \, r_1(x,y) \;\; \mathrm{s.t.} \;\; r_2(x,y) \geq b$ for some $b \in \mathbb{R}$ approaches the Pareto front. These limitations motivate an algorithm that optimizes all objectives *jointly*. Hence, we develop MOPO as an offline constrained optimization algorithm that recovers policies which lie near the true Pareto front as in Figure 1(b). Given this motivation, we now turn our attention to introducing notations and providing a formal problem statement.

**Problem Setting.** We define a finite set of contexts $\mathcal{X}$ and a finite action space $\mathcal{Y}$. A policy $\pi \in \Delta_{\mathcal{Y}}^{\mathcal{X}}$ defines a probability distribution over actions given a context, where $\Delta_{\mathcal{Y}}$ is the probability simplex over $\mathcal{Y}$. The policy is learned from human preferences, which are provided in a pairwise manner over actions. For each context $x \in \mathcal{X}$, two actions $y, y' \sim \mu(\cdot \,|\, x)$ are sampled from a behavior policy $\mu$, and a human annotator provides a preference signal indicating which action is preferred. We also let the contexts $x$ be sampled from a context distribution $\nu$, denote a vector by $\boldsymbol{v}$, let $\boldsymbol{v}_j$ to be the element at the $j^{th}$ dimension of $\boldsymbol{v}$, and let $[N]$ denote the set $\{1, \ldots, N\}$ for some $N \in \mathbb{N}$.

Typically, in single objective preference optimization, the preference for one generation over another is denoted as $y_w \succ y_l$, where $y_w$ and $y_l$ denote the preferred and dispreferred actions amongst $\{y, y'\}$ respectively. This true human preference takes the form $p(y \succ y' \,|\, x)$, the probability of $y$ being preferred to $y'$ knowing the context $x$. In our multi-objective preference setting, we extend this notation to $K$ objectives, wherein $p_k(y \succ y' \,|\, x)$ denotes the preferred and dis-preferred actions amongst $\{y, y'\}$ for $k^{th}$ objective with $k \in [K]$. Moreover, we also set the expected preference of a generation $y$ *over* a distribution $\mu$ knowing $x$ for the $k^{th}$ objective as $p_k(y \succ \mu \,|\, x) = \mathbb{E}_{y' \sim \mu(\cdot \,|\, x)}[p_k(y \succ y' \,|\, x)]$. We also let for any two policies $\pi, \mu \in \Delta_{\mathcal{Y}}^{\mathcal{X}}$ and a context distribution $\nu$ the total preference of policy $\pi$ to $\mu$ w.r.t. $k^{th}$ objective as $p_k^{\nu}(\pi \succ \mu) = \mathbb{E}_{x \sim \nu, y \sim \pi(.|x)}[p_k(y \succ \mu \,|\, x)]$. Without loss of generality and clarity of notation, we let $p_K(y \succ y' \,|\, x) \equiv p(y \succ y' \,|\, x)$ denote the preference for the $K^{th}$ (the primary) objective, and $\boldsymbol{q}(y \succ y' \,|\, x) \in [0,1]^{K-1}$ denotes the vector of preferences for the $K - 1$ (the secondary) remaining objectives, wherein the preferences are applied objective-wise i.e. $\boldsymbol{q}_k(y \succ y' \,|\, x) = p_k(y \succ y' \,|\, x)$ for $k \in [K-1]$. Following this notation, we also have the following definitions.

$$q(y \succ \mu \,|\, x) = \mathbb{E}_{y' \sim \mu(\cdot \,|\, x)}[q(y \succ y' \,|\, x)],$$

$$\text{and} \quad \boldsymbol{q}^{\nu}(\pi \succ \mu) = \mathbb{E}_{\substack{x \sim \nu \\ y \sim \pi(\cdot \,|\, x)}}[\boldsymbol{q}(y \succ \mu \,|\, x)].$$

**Pareto optimality.** In multi-objective preference optimization (MOPO), a policy that simultaneously optimizes all objectives does not exist. Thus, a set of non-dominated solutions is desired. We say policy $\pi$ is dominated by policy $\pi'$ when there is no objective under which $\pi'$ is *worse* than $\pi$, i.e., $p_k^{\nu}(\pi \succ \mu \,|\, x) \leq p_k^{\nu}(\pi' \succ \mu \,|\, x)$ for $\forall \, k \in [K]$. A policy $\pi$ is Pareto-optimal if and only if it is not dominated by any other policy. The Pareto set is composed of non-dominated solutions, denoted as $\Pi_P$. Overall, the goal of MOPO is to obtain an optimal policy in $\Pi_P$.

**Problem statement.** The goal is to propose a general solution for RLHF with multiple objectives, based on constrained optimization of a function of preferences. We propose this constrained optimization problem as maximizing a primary objective, and constraining the remaining objective values. To this end, we consider a reference policy $\pi_{\mathsf{ref}} \in \Delta_{\mathcal{Y}}^{\mathcal{X}}$, a real positive regularization parameter $\tau \in \mathbb{R}_+$, and let $\boldsymbol{b} \in [0,1]^{K-1}$. The concave constrained optimization problem (COP) for MOPO becomes,

$$
\begin{aligned}
\max_{\pi} \quad & \mathbb{E}_{\substack{x \sim \nu \\ y \sim \pi(\cdot \,|\, x), y' \sim \mu(\cdot \,|\, x)}} [p(y \succ y' \,|\, x)] - \tau \mathsf{KL}(\pi \,||\, \pi_{\mathsf{ref}}) \\
\mathrm{s.t.} \quad & \mathbb{E}_{\substack{x \sim \nu \\ y \sim \pi(\cdot \,|\, x), y' \sim \mu(\cdot \,|\, x)}} [\boldsymbol{q}(y \succ y' \,|\, x)] \geq \boldsymbol{b}.
\end{aligned}
\tag{1}
$$

See Definition A.1 for the definition of KL divergence. We now focus our attention on designing a MOPO algorithm to solve the COP problem above.

## 3. The MOPO Algorithm

First, to find an optimal policy that lies in $\Pi_P$, it is crucial to set proper constraint values $\boldsymbol{b}$ such that the solution of Problem (1) contributes to the

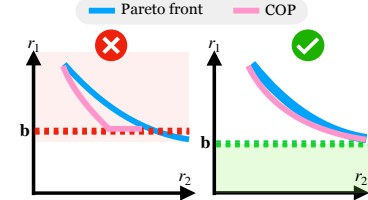

*Figure 2.* Illustration of how constraint threshold values affect COP solutions.

Pareto front. See Appendix A.2 and A.3 for the theoretical discussion, and Figure 2, which illustrates that if $\boldsymbol{b}$ is correctly initialized, then solving the COP problem yields solutions on the Pareto-front. In Section 3.1 we will propose a more practical method of specifying constraint values. For now, we focus on solving Problem (1). To deal with the optimization variable in expectation, we let the importance sampling ratio be $\rho(y) = \frac{\pi(y)}{\pi_{\mathsf{ref}}(y)}$. For this we assume that $\mathrm{Supp}(\pi) = \mathrm{Supp}(\pi_{\mathsf{ref}})$, and for clarity, we shall omit the dependency on context $x$ as the all results hold true for all $x \in \mathrm{Supp}(\nu)$. Then the final MOPO problem takes the form,

$$\max_{\rho} \underbrace{\mathbb{E}_{\substack{y\sim\pi_{\text{ref}}\\ y'\sim\mu}}[\rho(y)\,p(y\succ y')] - \tau\,\mathbb{E}_{y\sim\pi_{\text{ref}}}[\rho(y)\,\ln(\rho(y))]}_{\mathcal{F}(\rho)}$$

$$\text{s.t.}\quad \underbrace{\mathbb{E}_{\substack{y\sim\pi_{\text{ref}}\\ y'\sim\mu}}[\rho(y)\,\boldsymbol{q}(y\succ y')] \geq \boldsymbol{b}}_{\boldsymbol{\mathcal{G}}(\rho)},\tag{2}$$

which is a strictly concave optimization problem w.r.t. $\rho$. We then formulate the Lagrangian of the above `MOPO` problem. For some $\boldsymbol{\lambda} := \{\lambda_k\}_{k=1}^{K-1} \geq \boldsymbol{0}$, we have the Lagrangian as $\mathcal{L}(\rho, \boldsymbol{\lambda}) = \mathcal{F}(\rho) - \boldsymbol{\lambda}^T(\boldsymbol{b} - \boldsymbol{\mathcal{G}}(\rho))$. This leads to the following proposition.

**Proposition 3.1.** *The dual formulation of Problem* (2) *is given by,*

$$Dual^{\star} \triangleq \min_{\boldsymbol{\lambda}\geq 0}\max_{\rho} \mathcal{L}(\rho, \boldsymbol{\lambda}) = \min_{\boldsymbol{\lambda}\geq 0} \mathcal{L}(\rho_{\boldsymbol{\lambda}}^{\star}, \boldsymbol{\lambda})$$

$$= \min_{\boldsymbol{\lambda}\geq 0} \mathcal{F}(\rho_{\boldsymbol{\lambda}}^{\star}) - \boldsymbol{\lambda}^T(\boldsymbol{b} - \boldsymbol{\mathcal{G}}(\rho_{\boldsymbol{\lambda}}^{\star}))\quad, \text{where}, \quad \forall\, y\in\mathcal{Y},\tag{3}$$

$$\rho_{\boldsymbol{\lambda}}^{\star}(y) = \exp\left(\tau^{-1}\mathbb{E}_{y'\sim\mu}[p(y\succ y') + \boldsymbol{\lambda}^T\boldsymbol{q}(y\succ y')] - 1\right).$$

See Appendix A.5 for proof. The inner maximization in Equation (3) corresponds to computing an optimal policy (importance sampling ratio $\rho$) that maximizes scalarized preferences for the $K^{th}$ objective, while the outer minimization corresponds to balancing the penalty of suboptimal policy w.r.t. the other $(K-1)$ objectives: if the current policy ($\rho$) is under performing w.r.t. the $k^{th}$ objective, $\lambda_k$ increases so that the under performance is penalized more, and vice versa.

**Remark 3.2.** *Formulation in Problem* (3) *also connects to the use of barrier functions in optimization literature. For some $\sigma, s > 0$, consider the log barrier function for $z \in \mathbb{R}$,*

$$\mathcal{B}_{\sigma,s}(z) = \begin{cases} -\sigma\log(-z), & z \leq -s \\ \frac{\sigma}{s}z + (1-\log(s))\sigma, & z > -s \end{cases}.$$

*This relaxed log-barrier function can be used to construct an unconstrained Lagrangian with $\boldsymbol{\sigma} = \{\boldsymbol{\sigma}_k\}_{k=1}^{K-1} > \boldsymbol{0}$ as,*

$$\mathcal{L}_{LB}(\rho, \boldsymbol{\sigma}) = \mathcal{F}(\rho) - \sum_{k=1}^{K-1}\mathcal{B}_{\boldsymbol{\sigma}_k, \boldsymbol{\sigma}_k^2}(\boldsymbol{b}_k - \boldsymbol{\mathcal{G}}_k(\rho)).\tag{4}$$

*See Appendix A.4 for more details. Although comparable theoretically, we will see in Section 4 how formulation of Problem* (3) *is empirically superior to that of Problem* (4).

Returning to our discussion of Problem (3), we find that constraining the preference vector $\boldsymbol{q}(\cdot)$ naively can result in constraint violation when deployed to the real environment. This is due to the fact that empirical importance sampling weighted preferences $\widehat{\boldsymbol{q}}(\cdot)$ collected from a finite dataset inevitably have estimation error (see Figure 3). For the $K^{th}$ objective, preference estimation error may be tolerated as long as those estimates are useful as policy improvement signals, i.e., it is sufficient to maintain the relative order of preferences. For the remaining $(K-1)$ constrained objectives, Equation (3) instead relies on the estimated values directly. Hence, to make a policy robust against these

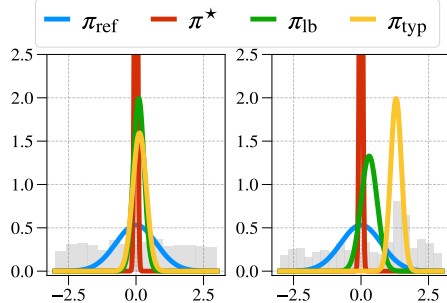

*Figure 3.* `MOPO`: Impact of preference distribution (gray) over output space with lower-magnitude outputs being preferred with probability 1. $\pi_{\text{lb}}$ constrains lower bound of $\boldsymbol{\mathcal{G}}(\rho)$, while $\pi_{\text{typ}}$ constrains $\boldsymbol{\mathcal{G}}(\rho)$ directly (typically).

estimation errors, we consider a constrained policy optimization scheme that instead constrains the *lower bound* of the preference estimates $\widehat{\boldsymbol{q}}(\cdot)$, i.e.,

$$\max_{\rho}\mathcal{F}(\rho)\quad \text{s.t.}\quad \text{LowerBound}(\boldsymbol{\mathcal{G}}(\rho)) \geq \boldsymbol{b}.$$

Then, the key question is how to estimate the lower bound of the $\rho$-weighted preference vector $\boldsymbol{q}(\cdot)$. One natural way is to exploit bootstrap confidence interval estimation (Efron & Tibshirani, 1994), by sampling bootstrap datasets $\mathcal{D}_i$ from $\mathcal{D}$ and constructing population statistics for confidence interval estimation. However, this procedure is computationally expensive. Instead, we take a different, computationally efficient approach. Specifically, we solve a constrained optimization problem for each objective $k \in [K-1]$. For a policy $\pi_k \in \Delta_{\mathcal{Y}}^{\mathcal{X}}$ and some $\epsilon \in \mathbb{R}_+$, the lower bound optimization problem of $\boldsymbol{\mathcal{G}}(\rho)$ becomes:

$$\min_{\pi_k}\mathbb{E}_{\substack{y\sim\pi_k\\ y'\sim\mu}}[\rho(y)\boldsymbol{q}_k(y\succ y')]\quad \text{s.t.}$$

$$\text{KL}(\pi_k\,||\,\pi_{\text{ref}}) \leq \epsilon\quad \text{and},\quad \sum_{y\in\mathcal{Y}}\pi_k(y) = 1.\tag{5}$$

In essence, we want to adversarially optimize a distribution $\pi_k$ so that it underestimates the preference objective $k$, and simultaneously, we enforce that this $\pi_k$ should not be perturbed too much from $\pi_{\text{ref}}$. As $\epsilon$ increases, the degree of underestimation of preference probabilities also increases. Now we simplify the constrained optimization problem into a single unconstrained problem as follows.

**Proposition 3.3.** *The optimal solution to Problem* (5) *can be obtained by solving the following optimization problem.*

$$\chi_k^{\star} = \operatorname*{argmax}_{\chi_k\geq 0}\ \mathcal{L}_k(\chi_k\,;\,\rho)$$

$$:= -\chi_k\ln\left(\mathbb{E}_{y\sim\pi_{\text{ref}}, y'\sim\mu}\left[\exp\left(-\chi_k^{-1}\rho(y)\boldsymbol{q}_k(y\succ y')\right)\right]\right) - \chi_k\epsilon$$

$$\text{with},\ \pi_k^{\star}(y) \propto \pi_{\text{ref}}(y)\underbrace{\exp\left(-(\chi_k^{\star})^{-1}\mathbb{E}_{y'\sim\mu}[\rho(y)\boldsymbol{q}_k(y\succ y')]\right)}_{w(y)\ (\textit{unnormalized weight})}.$$

See Appendix A.5 for proof. Note that each term in Equation (6) can be estimated using samples from the offline

dataset $\mathcal{D}$, thus it can be optimized in a fully offline manner. This procedure can be understood as computing the weights for each sample while adopting reweighting, that is, $\texttt{LowerBound}(\boldsymbol{\mathcal{G}}(\rho)) = \mathbb{E}_{y \sim \pi_{\text{ref}}, y' \sim \mu}[\widetilde{w}(y)\rho(y)\boldsymbol{q}_k(y \succ y')]$, where $\widetilde{w}(y)$ is normalized $w(y)$. Solving this unconstrained optimization problem and plugging it in the main $\texttt{Dual}^\star$ Problem (3) corresponds to the following iterative updates.

$$\chi^\star \leftarrow \operatorname*{argmax}_{\chi \geq 0} \sum_{k=1}^{K-1} \mathcal{L}_k(\chi_k; \rho_\lambda) \quad \text{and},$$

$$\lambda^\star \leftarrow \operatorname*{argmin}_{\lambda \geq 0} \mathcal{F}(\rho_\lambda) - \lambda^T(\boldsymbol{b} - \underbrace{\boldsymbol{\mathcal{L}}(\chi^\star; \rho_\lambda)}_{\texttt{LowerBound}(\boldsymbol{\mathcal{G}}(\rho))}), \tag{6}$$

where $\boldsymbol{\mathcal{L}}(\chi^\star; \rho_\lambda) = (\mathcal{L}_1(\chi^\star, \rho_\lambda), \ldots, \mathcal{L}_{K-1}(\chi^\star, \rho_\lambda))^T$. Compared to the original dual Problem (3), the additional maximization for $\chi$ is introduced to estimate the lower bound of preference probabilities for the constrained objectives. Once the optimal solution $\lambda^\star$ is computed, $\rho^\star_{\lambda^\star}(y) \equiv \rho^\star(y) = \pi^\star(y)/\pi_{\text{ref}}(y)$ is also derived from Equation (3).

**Policy Extraction.** The current procedure estimates the importance sampling ratio $\rho^\star(y)$ of the optimal policy, rather than directly obtaining the policy itself. Since the importance sampling ratio does not provide a direct way to sample an action, we need to extract the optimal policy $\pi^\star$ from $\rho^\star$ in order to select actions when deployed. For tabular cases, it is straightforward to obtain $\pi^\star(y) = (\pi_{\text{ref}}(y)\rho^\star(y)) / \left(\sum_{y' \in \mathcal{Y}} \pi_{\text{ref}}(y')\rho^\star(y')\right)$. However, the same method cannot directly be applied to large scale optimization problems due to the intractability of computing the normalization constant. For such cases, we instead extract the policy using importance-weighted behavioral cloning by solving the following problem:

$$\max_\pi \mathbb{E}_{y \sim \pi^\star}[\log(\pi(y))] = \max_\pi \mathbb{E}_{y \sim \pi_{\text{ref}}}[\rho^\star(y) \log(\pi(y))], \tag{7}$$

which maximizes the log-likelihood of actions to be selected by the optimal policy $\pi^\star$. We now turn our attention to the practical implementation of $\texttt{MOPO}$.

### 3.1. Practical Algorithm with Function Approximation

For this section we consider the practical implementation of $\texttt{MOPO}$, using a given offline dataset of preferences $\mathcal{D}$. First, we discuss the function approximations used to parameterize the optimization variables, and then we discuss how to implement the procedure discussed above.

**Function Approximations.** We let the optimization variables $\lambda, \chi \in \mathbb{R}_+^{K-1}$, and the policy $\pi_\psi$ to be parameterized by $\psi$. The parameter $\chi = \{\chi_k\}_{k=1}^{K-1}$ is trained by minimizing the following loss:

$$\min_{\chi \geq 0} \sum_{k=1}^{K-1} \left[\chi_k \ln \left(\mathbb{E}_{y \sim \pi_{\text{ref}}, y' \sim \mu}\left[\exp\left(\chi_k^{-1}\rho(y)\boldsymbol{q}_k(y \succ y')\right)\right]\right) + \chi_k \epsilon\right].$$

Since this involves a logarithm outside of the expectation, to overcome bias we use mini-batch approximation for computational efficiency. The empirical form is then given by:

$$\min_{\chi \geq 0} J(\chi; \rho) := \mathbb{E}_{\text{batch}(\mathcal{D}) \sim \mathcal{D}}\left[\sum_{k=1}^{K-1} \left[\chi_k \ln\left(\mathbb{E}_{y, y' \sim \text{batch}(\mathcal{D})}\left[\exp\left(\chi_k^{-1}\rho(y)\boldsymbol{q}_k(y \succ y')\right)\right]\right) + \chi_k \epsilon\right]\right]. \tag{8}$$

Finally, following the discussion before, $\lambda$ and the policy parameterized by $\psi$ are optimized by:

$$\min_{\lambda \geq 0} J(\lambda; \chi) := \mathcal{F}(\rho) - \lambda^T(\boldsymbol{b} - J(\chi; \rho)) \quad \text{and},$$

$$\min_\psi J_\rho(\pi_\psi) := -\mathbb{E}_{y \sim \pi_{\text{ref}}}[\rho(y)\log(\pi_\psi(y))], \tag{9}$$

where all variables are optimized jointly. For the empirical derivation of $\texttt{MOPO}$ given a *fixed* offline dataset of preferences, please see Appendix A.6, where we discuss its practical implementation. However, two caveats still remain.

**Lagged reference policy.** The KL regularizer in Equation (2) keeps $\pi_\psi$ close to a fixed reference $\pi_{\text{ref}}$. Because successive iterates move toward the Pareto front, it is advantageous to regularize against a *stronger* policy than the initial prior. Analogous to target networks in Q-learning (Mnih et al., 2013) and recent self-improvement loops for LLMs (Chen et al., 2024; Pang et al., 2024), we update the reference every $t_0$ steps: $\pi_{\text{ref}} \leftarrow \pi_\psi^{(t-t_0)}$. All expectations in Equation (25) are then reweighted by the ratio $\rho_{\text{lag,ref}}(y) = \pi_\psi^{(t-t_0)}(y)/\pi_{\text{ref}}(y)$ for all $y \in \mathcal{Y}$, requiring no further data collection. We find that this leads to more stable optimization and consistent progression to the Pareto front.

**Adaptive constraint schedule.** In practice, exact values of constraint thresholds $\boldsymbol{b}$ are unknown. We therefore, after every $t_0$ steps, set the constraint vector *only* from the policy of the previously optimized iterates: $\boldsymbol{b} = \beta^\top \boldsymbol{\mathcal{G}}\left(\rho^{(t-t_0)}\right)$ for some hyperparameter $\beta \in (0, 1)^{K-1}$ and $\rho^{(t-t_0)}(\cdot) = \pi_\psi^{(t-t_0)}(\cdot)/\pi_{\text{ref}}(\cdot)$ is the importance sampling ratio. This retains the theoretical lower bound interpretation of the constraints while avoiding a global search across thresholds.

Summarizing the above discussion gives the final *Multi-Objective Preference Optimization* ($\texttt{MOPO}$) algorithm, shown in Algorithm 1. At each step we maximize the primary preference objective subject to the time-varying lower bounds $\boldsymbol{b}$, while penalizing divergence from the current reference policy. The result is a scalable, offline algorithm that steadily advances toward Pareto-optimal solutions.

## 4. Empirical Results

We conducted extensive experimental evaluation on the relative empirical performance of the $\texttt{MOPO}$ algorithm to arrive at the following conclusions: (i) $\texttt{MOPO}$ exactly recovers the optimal policy under some canonical ordering of preferences, (ii) in contrast to several DPO-style baselines (Gheshlaghi Azar et al., 2024), $\texttt{MOPO}$ does not overfit to the preference dataset and preserves performance on held-out comparisons, and (iii) it performs better or nearly as well as baseline algorithms when evaluated on LLMs on real data.

**Algorithm 1** Multi-objective Preference Optimization

1: **Input:** Dataset $\mathcal{D}$, batch size $M$, learning rate $\eta$, epochs $T$, lag $t_0$, relaxation parameter $\boldsymbol{\beta}$.
2: Initialize parameters $\boldsymbol{\lambda}^{(0)}, \boldsymbol{\chi}^{(0)}, \psi^{(0)}, \boldsymbol{b}, \rho_{\boldsymbol{\lambda}^{(0)}}(\cdot)$.
3: **for** $t = 1, 2, \ldots, T$ **do**
4:     Sample $M$ mini-batches from $\mathcal{D}$.
5:     $\boldsymbol{\chi}^{(t)} = [\boldsymbol{\chi}^{(t-1)} - \eta \nabla_{\boldsymbol{\chi}} J(\boldsymbol{\chi}; \rho_{\boldsymbol{\lambda}^{(t-1)}})]_+$ using Equation (8).
6:     $\boldsymbol{\lambda}^{(t)} = [\boldsymbol{\lambda}^{(t-1)} - \eta \nabla_{\boldsymbol{\lambda}} J(\boldsymbol{\lambda}; \boldsymbol{\chi}^{(t)})]_+$ using Equation (9).
7:     Compute $\rho_{\boldsymbol{\lambda}^{(t)}}(\cdot)$ using Equation (3).
8:     Update policy $\psi^{(t)} = \psi^{(t-1)} - \eta \nabla_\psi J_{\rho_{\boldsymbol{\lambda}^{(t)}}}(\pi_\psi)$ using Equation (9).
9:     **if** $t \bmod t_0 = 0$ **then**
10:         $\boldsymbol{b} = \boldsymbol{\beta}^\top \boldsymbol{\mathcal{G}}\left(\rho^{(t-t_0)}\right)$
11:         $\pi_{\mathsf{ref}} \leftarrow \pi_\psi^{(t-t_0)}$
12:     **end if**
13: **end for**
14: **Output:** Optimal policy $\pi_\psi^{(T)}$.

## 4.1. Synthetic Sanity Check

Here we verify if `MOPO` solves the optimization problem exactly when ground-truth optimum is known. We consider a two-objective, context-free bandit setting with discrete action set $\mathcal{Y} = \{y_1, y_2, y_3\}$ and a uniform reference policy $\pi_{\mathsf{ref}}$. Training data $\mathcal{D} = \{(y_i, y_i', \mathbb{I}(y_i, y_i'))\}_{i=1}^N$, where $\mathbb{I}(\cdot, \cdot) \in \{0, 1\}^2$ is the preference indicator vector, i.e., $\mathbb{I}_k(y, y') = 1$ if $y \succ y'$ for the $k^{\text{th}}$ objective, and 0 otherwise for $k \in \{1, 2\}$, of size $N$ consists of one of three canonical preference structures respectively: total order, partial order, and unobserved preferences.

$$\mathcal{D}_1 = \{(y_1, y_2, (1,1)), (y_2, y_3, (1,1)), (y_1, y_3, (1,1))\}$$
$$\mathcal{D}_2 = \{(y_1, y_2, (1,1)), (y_1, y_3, (1,0)), (y_2, y_3, (0,0))\}$$
$$\mathcal{D}_3 = \{(y_1, y_2, (1,1)), (y_2, y_1, (0,1))\}$$

**Learning Protocol.** Mini-batches are drawn uniformly *with replacement* from each of $\mathcal{D}_j$ for $j \in \{1, 2, 3\}$ and optimized for 20k steps using Adam (Kingma & Ba, 2014) with a learning rate of 0.015 and batch size 12. Policy is encoded simply as $\pi_\psi(y_i) = \mathrm{softmax}(\boldsymbol{\psi})_i$ using $\boldsymbol{\psi} \in \mathbb{R}^3$.

**Results.** Learned action probabilities for each $\mathcal{D}_j$ (column-wise) are seen in Figure 4. For $\mathcal{D}_1$ the policy converges to the Condorcet winner $y_1$. For $\mathcal{D}_2$, in which $y_1$ and $y_3$ are undominated, `MOPO` assigns them equal probability. Finally, with the inconsistent set $\mathcal{D}_3$, `MOPO` successively down-weights the unobserved action $y_3$ as $\tau$ decreases. Across all cases, increasing $\tau$ interpolates between $\pi_{\mathsf{ref}}$ and the optimal policy, confirming controlled regularization.

## 4.2. Experiments on Text Generation Tasks

Having established the validity of alignment using `MOPO`, we now aim to evaluate the performance of the `MOPO` algorithm on text generation tasks that involve diverse rewards.

**Baselines.** We consider (i) Rewards-in-Context (RiC) (Yang et al., 2024) with in-context rewards and human preferences (this is a state-of-the-art baseline that outperforms Rewarded Soups (Rame et al., 2023) and MORLHF (Li et al., 2020)), (ii) Preference-aware Autoregressive Reward Model (PARM) (Lin et al., 2025), a single unified model trained across all preference dimensions, (iii) Multi-objective DPO (MODPO) (Zhou et al., 2023), which is a more nuanced version of scalarizing multiple reward models into one (this corresponds to "DPO on $w^T r$" in Figure 1), and (iv) DPO on $\mathcal{D}_J$, which trains DPO on context-output pairs which are preferred under both reward models similar to our discussion in Section 2. We also compare two verions of `MOPO`: (i) using the log-barrier formulation of Remark 3.2 called `MOPO-LB`, and (ii) using the Lagrangian called `MOPO-Lag`.

**Benchmarks and Training.** All methods are evaluated based on the quality of their empirical Pareto fronts on the Helpful Assistant task (Bai et al., 2022) and the Reddit Summarization task (Stiennon et al., 2020). The Helpful Assistant task uses the HH-RLHF dataset containing 160k prompts with human preference annotations. Evaluation is performed using three HuggingFace reward models –'helpful', 'harmless', and 'humor' – which score responses from different perspectives (Wolf et al., 2019). The Reddit Summary task consists of 14.9k posts and their summaries. We consider three reward models: 'pref1' and 'no-hallucinate', which evaluate human preference for summaries, and a 'faithful' reward that measures the faithfulness of the summary to the original post. We apply these benchmarks to publicly available 7B-level models that have shown strong helpfulness scores (Gao et al., 2024; Dubois et al., 2023). In Figure 5, even though we observe that Zephyr-7b-beta (Tunstall et al., 2023), an open-source model fine-tuned over Mistral-7B-v0.1 (Jiang et al., 2023), Pareto dominates other models when aligned using `MOPO`, we conduct experiments on various models to show generalizability of `MOPO`. All likelihood maximization problems use parameter efficient fine-tuning with LoRA (Hu et al., 2022a) for 10k steps with a batch size of 8. LoRA is applied to the shared transformer backbone, and is optimized together with the policy parameters. See Appendix A.7 for more details.

**Evaluation protocol.** For both datasets, we uniformly sample 3k prompts from the test sets and cluster them into $j = 6$ clusters, and compute the average reward for each objective across cluster groups (Mukherjee et al., 2024; Yang et al., 2024). Performance is measured by comparing the resulting multi-objective reward values.

See Table 1 for results over 5 independent runs. Each cell represents reward tuples along with 1 standard deviation of models when aligned with baselines and trained on the corresponding preference dataset. For `MOPO`, we let the primary objective be $r_1$ and constrain preferences w.r.t. $r_2$. For RiC, PARM, and MODPO, values with highest rewards across

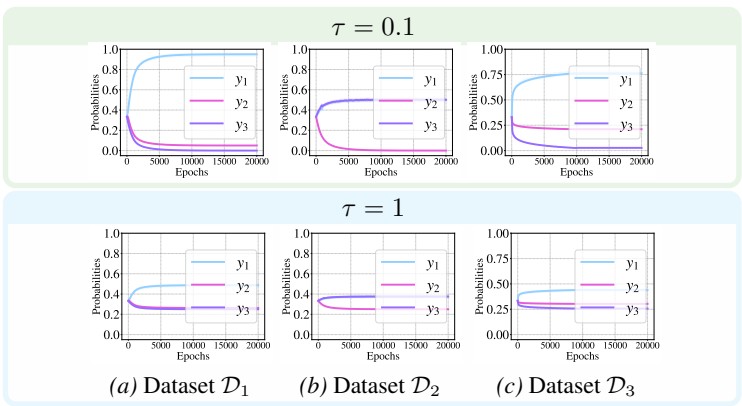

*Figure 4.* Learning curves of action probabilities of `MOPO` on various dataset types (read column-wise).

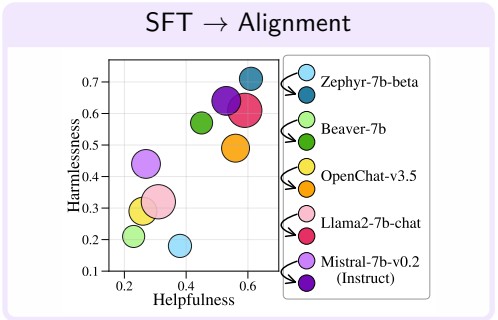

*Figure 5.* Open-sourced models before (pastel) and after (saturated) alignment using `MOPO`. Circle size represents approximate training data size and annotation cost.

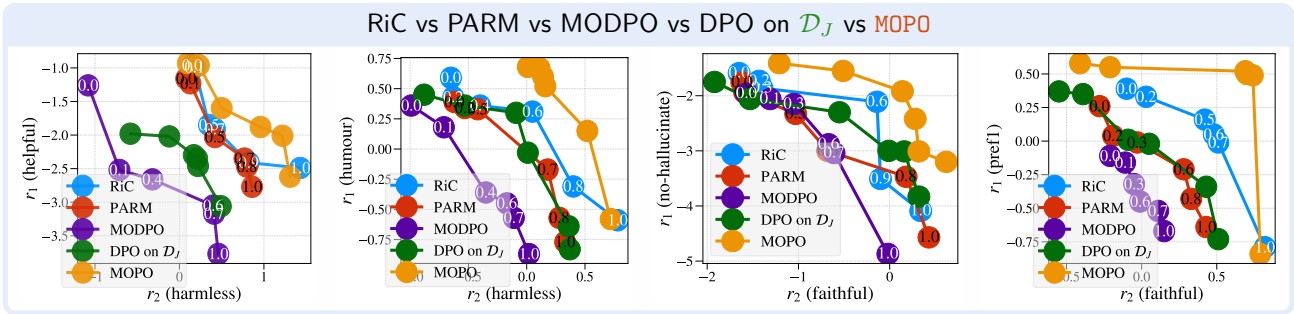

*Figure 6.* Empirical Pareto fronts of phi-1.5 when aligned using baselines on the Helpful Assistant and Reddit Summary tasks.

inference preference vector inputs are shown. According to the results, `MOPO` achieves substantial alignment improvement for most objectives and policy models. From a model scale perspective, `MOPO` is able to scale from tiny-LLMs (phi-1.5) to larger LLMs as well (Llama-3.1-8B). In Figure 6 we use phi-1.5 as the SFT model for alignment tasks to empirically visualize Pareto fronts of all baselines. Each point represents the average rewards evaluated on sampled cluster groups from the test set. For `MOPO`, we vary the constraint threshold w.r.t. $r_2$ to obtain points on the empirical front. For RiC and PARM, the numbers at the centers of the markers indicate the (normalized) preference for $r_2$ in each cluster that achieves the highest reward within that cluster. It is clear from the results that `MOPO` consistently approximates the Pareto front as well as or better than PARM and RiC.

To assess the scalability of `MOPO`, we aim to optimize three objectives in the Helpful Assistant task, i.e., 'harmless', 'helpful', and 'humour'. We use Zephyr-7b-beta as our base SFT model. For easy interpretation, we sample 2k prompts from the test set and plot

the average rewards. The results in Table 2 reveal that RLHF (Yang et al., 2024), when optimized for a single reward (see Appendix A.7 for the problem formulation), achieves high performance on the targeted reward but degrades substantially on the remaining objectives. In contrast, multi-objective algorithms yield more uniform performance across all rewards, with `MOPO` achieving the most balanced trade-offs. The results demonstrate the effectiveness of `MOPO` in scaling to more than two objectives.

*Table 2.* Three objective alignment for Helpful Assistant task with normalized rewards.

|         | helpful | humour | harmless |
|---------|---------|--------|----------|
| RLHF-r1 | 0.76    | -0.42  | -0.23    |
| RLHF-r2 | -0.81   | 0.53   | -0.40    |
| RLHF-r3 | -0.79   | -0.92  | 0.42     |
| RiC     | 0.25    | 0.15   | 0.11     |
| PARM    | 0.31    | 0.17   | 0.23     |
| MODPO   | 0.04    | -0.09  | 0.08     |
| DPO on $\mathcal{D}_J$ | 0.18 | 0.09 | 0.11  |
| MOPO-LB | 0.30    | 0.19   | 0.18     |
| MOPO-Lag| 0.39    | 0.22   | 0.17     |

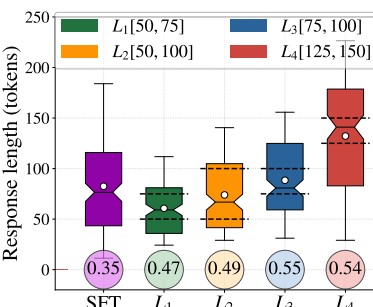

*Figure 7.* Response lengths along with helpfulness scores in circles, where notches indicate medians, boxes show $\pm 25\%$ quantiles, white circles mark the means, and the black dashed lines depict the constraints imposed.

*Table 1.* Results of the Helpful Assistant task and the Reddit Summary task with normalized rewards. Color intensity reflects per-column magnitude (red = low, green = high).

| Model | Algorithm | helpful-harmless $(r_1, r_2)$ | | humour-harmless $(r_1, r_2)$ | | no_hallucinate-faithful $(r_1, r_2)$ | | pref1-faithful $(r_1, r_2)$ | |
|---|---|---|---|---|---|---|---|---|---|
| **phi-1.5** (Li et al., 2023) | RiC | ( $0.23_{\pm0.03}$ , | $0.21_{\pm0.03}$ ) | ( $0.31_{\pm0.04}$ , | $0.27_{\pm0.03}$ ) | ( $0.40_{\pm0.06}$ , | $0.37_{\pm0.05}$ ) | ( $0.39_{\pm0.05}$ , | $0.36_{\pm0.07}$ ) |
| | PARM | ( $0.28_{\pm0.02}$ , | $0.24_{\pm0.05}$ ) | ( $0.33_{\pm0.05}$ , | $0.32_{\pm0.04}$ ) | ( $0.39_{\pm0.05}$ , | $0.41_{\pm0.05}$ ) | ( $0.47_{\pm0.07}$ , | $0.49_{\pm0.06}$ ) |
| | MODPO | ( $0.21_{\pm0.04}$ , | $0.20_{\pm0.04}$ ) | ( $0.25_{\pm0.04}$ , | $0.24_{\pm0.06}$ ) | ( $0.34_{\pm0.07}$ , | $0.35_{\pm0.04}$ ) | ( $0.31_{\pm0.06}$ , | $0.30_{\pm0.04}$ ) |
| | DPO on $\mathcal{D}_J$ | ( $0.22_{\pm0.03}$ , | $0.19_{\pm0.03}$ ) | ( $0.27_{\pm0.05}$ , | $0.25_{\pm0.04}$ ) | ( $0.35_{\pm0.03}$ , | $0.35_{\pm0.05}$ ) | ( $0.31_{\pm0.04}$ , | $0.34_{\pm0.03}$ ) |
| | MOPO-LB | ( $0.27_{\pm0.04}$ , | $0.24_{\pm0.04}$ ) | ( $0.34_{\pm0.06}$ , | $0.35_{\pm0.05}$ ) | ( $0.46_{\pm0.07}$ , | $0.40_{\pm0.04}$ ) | ( $0.44_{\pm0.03}$ , | $0.47_{\pm0.05}$ ) |
| | MOPO-Lag | ( $0.29_{\pm0.05}$ , | $0.26_{\pm0.03}$ ) | ( $0.37_{\pm0.06}$ , | $0.36_{\pm0.06}$ ) | ( $0.42_{\pm0.04}$ , | $0.39_{\pm0.05}$ ) | ( $0.47_{\pm0.08}$ , | $0.45_{\pm0.06}$ ) |
| **OpenChat-v3.5** (Wang et al., 2023) | RiC | ( $0.40_{\pm0.04}$ , | $0.33_{\pm0.03}$ ) | ( $0.37_{\pm0.03}$ , | $0.40_{\pm0.04}$ ) | ( $0.55_{\pm0.05}$ , | $0.43_{\pm0.04}$ ) | ( $0.40_{\pm0.04}$ , | $0.43_{\pm0.03}$ ) |
| | PARM | ( $0.38_{\pm0.04}$ , | $0.37_{\pm0.06}$ ) | ( $0.36_{\pm0.04}$ , | $0.39_{\pm0.04}$ ) | ( $0.60_{\pm0.04}$ , | $0.47_{\pm0.02}$ ) | ( $0.39_{\pm0.04}$ , | $0.40_{\pm0.05}$ ) |
| | MODPO | ( $0.35_{\pm0.05}$ , | $0.33_{\pm0.04}$ ) | ( $0.39_{\pm0.04}$ , | $0.37_{\pm0.05}$ ) | ( $0.51_{\pm0.03}$ , | $0.48_{\pm0.04}$ ) | ( $0.41_{\pm0.04}$ , | $0.39_{\pm0.02}$ ) |
| | DPO on $\mathcal{D}_J$ | ( $0.28_{\pm0.02}$ , | $0.26_{\pm0.02}$ ) | ( $0.36_{\pm0.03}$ , | $0.35_{\pm0.02}$ ) | ( $0.52_{\pm0.04}$ , | $0.47_{\pm0.06}$ ) | ( $0.35_{\pm0.01}$ , | $0.37_{\pm0.03}$ ) |
| | MOPO-LB | ( $0.41_{\pm0.04}$ , | $0.39_{\pm0.03}$ ) | ( $0.40_{\pm0.05}$ , | $0.39_{\pm0.05}$ ) | ( $0.63_{\pm0.02}$ , | $0.50_{\pm0.03}$ ) | ( $0.46_{\pm0.02}$ , | $0.44_{\pm0.03}$ ) |
| | MOPO-Lag | ( $0.43_{\pm0.05}$ , | $0.41_{\pm0.04}$ ) | ( $0.42_{\pm0.04}$ , | $0.40_{\pm0.04}$ ) | ( $0.61_{\pm0.01}$ , | $0.49_{\pm0.04}$ ) | ( $0.44_{\pm0.04}$ , | $0.42_{\pm0.01}$ ) |
| **Llama-3.1-8B** (Grattafiori et al., 2024) | RiC | ( $0.41_{\pm0.05}$ , | $0.47_{\pm0.03}$ ) | ( $0.43_{\pm0.04}$ , | $0.41_{\pm0.02}$ ) | ( $0.40_{\pm0.03}$ , | $0.42_{\pm0.05}$ ) | ( $0.40_{\pm0.05}$ , | $0.41_{\pm0.03}$ ) |
| | PARM | ( $0.40_{\pm0.01}$ , | $0.42_{\pm0.04}$ ) | ( $0.35_{\pm0.03}$ , | $0.44_{\pm0.01}$ ) | ( $0.44_{\pm0.04}$ , | $0.55_{\pm0.01}$ ) | ( $0.49_{\pm0.03}$ , | $0.48_{\pm0.04}$ ) |
| | MODPO | ( $0.32_{\pm0.04}$ , | $0.33_{\pm0.03}$ ) | ( $0.37_{\pm0.05}$ , | $0.45_{\pm0.04}$ ) | ( $0.46_{\pm0.02}$ , | $0.46_{\pm0.03}$ ) | ( $0.48_{\pm0.04}$ , | $0.46_{\pm0.06}$ ) |
| | DPO on $\mathcal{D}_J$ | ( $0.31_{\pm0.03}$ , | $0.30_{\pm0.04}$ ) | ( $0.35_{\pm0.04}$ , | $0.40_{\pm0.04}$ ) | ( $0.39_{\pm0.04}$ , | $0.40_{\pm0.03}$ ) | ( $0.39_{\pm0.01}$ , | $0.39_{\pm0.03}$ ) |
| | MOPO-LB | ( $0.45_{\pm0.02}$ , | $0.43_{\pm0.05}$ ) | ( $0.42_{\pm0.05}$ , | $0.50_{\pm0.02}$ ) | ( $0.51_{\pm0.03}$ , | $0.49_{\pm0.01}$ ) | ( $0.52_{\pm0.03}$ , | $0.50_{\pm0.05}$ ) |
| | MOPO-Lag | ( $0.48_{\pm0.02}$ , | $0.46_{\pm0.03}$ ) | ( $0.45_{\pm0.03}$ , | $0.53_{\pm0.05}$ ) | ( $0.54_{\pm0.05}$ , | $0.52_{\pm0.02}$ ) | ( $0.50_{\pm0.05}$ , | $0.49_{\pm0.02}$ ) |
| **Mistral-7b-v0.2 (Instruct)** (Jiang et al., 2023) | RiC | ( $0.44_{\pm0.04}$ , | $0.41_{\pm0.03}$ ) | ( $0.43_{\pm0.05}$ , | $0.44_{\pm0.02}$ ) | ( $0.45_{\pm0.04}$ , | $0.41_{\pm0.03}$ ) | ( $0.46_{\pm0.02}$ , | $0.44_{\pm0.05}$ ) |
| | PARM | ( $0.48_{\pm0.03}$ , | $0.47_{\pm0.01}$ ) | ( $0.41_{\pm0.03}$ , | $0.46_{\pm0.04}$ ) | ( $0.43_{\pm0.02}$ , | $0.42_{\pm0.02}$ ) | ( $0.43_{\pm0.05}$ , | $0.42_{\pm0.04}$ ) |
| | MODPO | ( $0.41_{\pm0.04}$ , | $0.39_{\pm0.04}$ ) | ( $0.40_{\pm0.02}$ , | $0.42_{\pm0.02}$ ) | ( $0.43_{\pm0.03}$ , | $0.43_{\pm0.05}$ ) | ( $0.37_{\pm0.04}$ , | $0.38_{\pm0.03}$ ) |
| | DPO on $\mathcal{D}_J$ | ( $0.32_{\pm0.01}$ , | $0.30_{\pm0.02}$ ) | ( $0.36_{\pm0.04}$ , | $0.39_{\pm0.03}$ ) | ( $0.39_{\pm0.05}$ , | $0.40_{\pm0.01}$ ) | ( $0.36_{\pm0.03}$ , | $0.35_{\pm0.04}$ ) |
| | MOPO-LB | ( $0.44_{\pm0.02}$ , | $0.45_{\pm0.04}$ ) | ( $0.49_{\pm0.04}$ , | $0.48_{\pm0.05}$ ) | ( $0.47_{\pm0.04}$ , | $0.45_{\pm0.03}$ ) | ( $0.45_{\pm0.04}$ , | $0.43_{\pm0.03}$ ) |
| | MOPO-Lag | ( $0.45_{\pm0.04}$ , | $0.47_{\pm0.03}$ ) | ( $0.49_{\pm0.02}$ , | $0.47_{\pm0.04}$ ) | ( $0.48_{\pm0.02}$ , | $0.46_{\pm0.02}$ ) | ( $0.45_{\pm0.02}$ , | $0.42_{\pm0.02}$ ) |
| **Zephyr-7b-beta** (Tunstall et al., 2023) | RiC | ( $0.43_{\pm0.02}$ , | $0.46_{\pm0.03}$ ) | ( $0.45_{\pm0.02}$ , | $0.47_{\pm0.03}$ ) | ( $0.50_{\pm0.03}$ , | $0.48_{\pm0.02}$ ) | ( $0.48_{\pm0.02}$ , | $0.44_{\pm0.04}$ ) |
| | PARM | ( $0.45_{\pm0.01}$ , | $0.48_{\pm0.02}$ ) | ( $0.48_{\pm0.04}$ , | $0.50_{\pm0.02}$ ) | ( $0.56_{\pm0.04}$ , | $0.53_{\pm0.03}$ ) | ( $0.52_{\pm0.01}$ , | $0.46_{\pm0.03}$ ) |
| | MODPO | ( $0.35_{\pm0.01}$ , | $0.33_{\pm0.04}$ ) | ( $0.40_{\pm0.03}$ , | $0.40_{\pm0.05}$ ) | ( $0.43_{\pm0.01}$ , | $0.44_{\pm0.02}$ ) | ( $0.39_{\pm0.03}$ , | $0.40_{\pm0.02}$ ) |
| | DPO on $\mathcal{D}_J$ | ( $0.34_{\pm0.03}$ , | $0.38_{\pm0.02}$ ) | ( $0.38_{\pm0.01}$ , | $0.41_{\pm0.03}$ ) | ( $0.41_{\pm0.02}$ , | $0.43_{\pm0.05}$ ) | ( $0.36_{\pm0.04}$ , | $0.36_{\pm0.05}$ ) |
| | MOPO-LB | ( $0.51_{\pm0.04}$ , | $0.52_{\pm0.04}$ ) | ( $0.49_{\pm0.02}$ , | $0.48_{\pm0.04}$ ) | ( $0.55_{\pm0.03}$ , | $0.51_{\pm0.01}$ ) | ( $0.48_{\pm0.02}$ , | $0.43_{\pm0.03}$ ) |
| | MOPO-Lag | ( $0.53_{\pm0.02}$ , | $0.51_{\pm0.03}$ ) | ( $0.52_{\pm0.03}$ , | $0.53_{\pm0.04}$ ) | ( $0.60_{\pm0.02}$ , | $0.55_{\pm0.03}$ ) | ( $0.50_{\pm0.04}$ , | $0.44_{\pm0.02}$ ) |

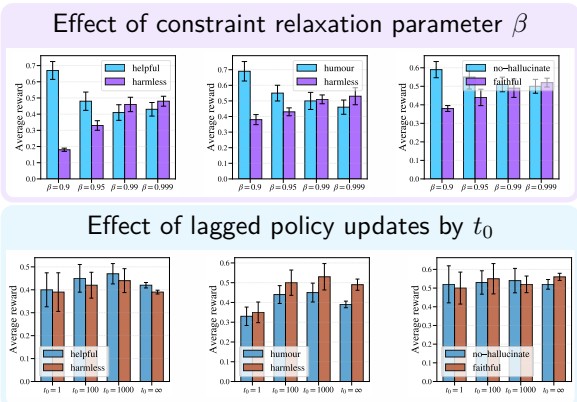

*Figure 8.* Normalized rewards of Llama-3.1-8B aligned using MOPO under varying optimization parameters on the Helpful Assistant and Reddit Summary tasks.

### 4.3. Secondary Evaluation

We now take a deeper dive into MOPO's performance and discuss its dependence on various factors.

**Verbosity-constrained tasks.** Consider tasks in which the lengths of LLM responses need to be contained in the range $[l_{\text{low}}, l_{\text{high}}]$ to control verbosity (while maintaining helpfulness); for example, in summarization tasks (Makino et al.,

2019). In this case, the natural choice for reward functions is to directly constrain the rewards $r_1(y|\cdot) = |y| \geq l_{\text{low}}$ and $r_2(y|\cdot) = -|y| \geq -l_{\text{high}}$. We illustrate the distributions of the generated response lengths (in tokens) by Zephyr-7b-beta aligned using MOPO, and report the corresponding helpfulness scores in Figure 7. We observe that the mean response lengths are in the required range in each case, satisfying the imposed constraints while improving helpfulness.

**Sensitivity analysis.** Updates in the MOPO algorithm are governed by optimization hyperparameters such as the constraint relaxation factor $\beta$ and the lag interval $t_0$. In practice, robustness to these hyperparameters is desirable, as their tuning can significantly affect performance. Figure 8 shows that MOPO remains stable and effective even under suboptimal choices of $\beta$ and $t_0$. See Appendix A.8 for full results.

## 5. Conclusion

In this paper, we introduced MOPO, an offline, multi-objective constrained optimization algorithm that learns from preference data and maximizes a primary objective while enforcing tunable lower bound constraints on secondary objectives. On synthetic benchmarks MOPO accurately recovers the true Pareto front. Experiments on real-

world datasets show that `MOPO` matches or surpasses base-lines, and ablation studies prove robustness to hyperparameters. An important future direction is to develop a rigorous theoretical analysis of `MOPO`.

## Impact Statement

This paper presents work whose goal is to advance the field of Machine Learning. There are many potential societal consequences of our work, none which we feel must be specifically highlighted here.

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

# A. Appendix

**Definition A.1.** *For any two policies $P, Q \in \Delta_{\mathcal{Y}}^{\mathcal{X}}$ such that* `Supp(P) = Supp(Q)`*, their* KL *divergence is defined as:*

$$\mathsf{KL}\left(P \,\|\, Q\right) = \mathbb{E}_{\substack{x \sim \nu \\ y \sim P(\cdot|x)}} \left[ \log\left( \frac{P(y \,|\, x)}{Q(y \,|\, x)} \right) \right] .$$

## A.1. A Detailed Motivating Example

In this example, we empirically demonstrate the necessity of *principled* multi-objective optimization methods that account for multi-dimensional preferences. We benchmark various approaches for multi-objective alignment and show that existing state-of-the-art techniques consistently fail to reach the Pareto front. To ensure clarity, we conduct experiments on synthetic datasets where the true Pareto front is known, allowing for precise evaluation of alignment quality.

Now define the input space $\mathcal{X} := \mathcal{U}[0,1]$ and output space $\mathcal{Y} := \mathcal{U}[0,1]$. For input $x \in \mathcal{X}$ and outputs $y, y' \in \mathcal{Y}$, the Bradley-Terry preference model $\mathsf{BT}(r(\cdot))$ (Bradley & Terry, 1952) w.r.t. to a reward model $r(\cdot)$ provides preference $z$ as,

$$\Pr(z = y \mid x, y, y') = r(x, y) / \left( r(x, y) + r(x, y') \right) \tag{10}$$

**Reward models.** For $(x, y) \in \mathcal{X} \times \mathcal{Y}$, we consider two pairs of reward functions $r : \mathcal{X} \times \mathcal{Y} \to \mathbb{R}$:

$$r_1^A(x, y) = e^x + \sqrt{y} - y \quad \text{and} \quad r_2^A(x, y) = -\sin(x) - y^2.$$
$$r_1^B(x, y) = (x + y)^2 \quad \text{and} \quad r_2^B(x, y) = \log(\frac{1+x}{1+y}). \tag{11}$$

**Dataset construction.** For arbitrary $N \in \mathbb{N}$, generate $x_i, y_i, y_i' \sim \mathcal{U}[0,1]$ for $i \in [N]$. Let $z_i^{(1)} \sim \mathsf{BT}(r_1)$ and $z_i^{(2)} \sim \mathsf{BT}(r_2)$ with $z_i^{(1)}, z_i^{(2)} \in \{y_i, y_i'\}$. We now construct four datasets: (i) $\mathcal{D}_1 = \{(x_i, y_i, y_i', z_i^{(1)})\}_{i=1}^N$ incorporating preferences w.r.t. reward model $r_1(\cdot)$ *only*, (ii) $\mathcal{D}_2 = \{(x_i, y_i, y_i', z_i^{(2)})\}_{i=1}^N$ incorporating preferences w.r.t. reward model $r_2(\cdot)$ *only*, (iii) $\mathcal{D}_J = \{(x_i, y_i, y_i', z) : z = z_i^{(1)} = z_i^{(2)}\}_{i=1}^N$ incorporating preferences only if they are consistent with reward models $r_1(\cdot)$ *and* $r_2(\cdot)$, and (iv) $\mathcal{D}_C = \{(x_i, y_i, y_i', z_i^{(C)}) : z_i^{(C)} \sim \mathsf{BT}(wr_1 + (1-w)r_2)\}_{i=1}^N$ for some $w \in [0,1]$, which incorporates preferences based on some convex weighting of both reward models.

Another approach of solving the multi-objective preference problem is a constrained optimization approach COP, where we can solve for the optimal policy as $\pi_{\mathrm{COP}}(x) = \mathrm{argmax}_y \; r_1(x, y) \; \mathtt{s.t.} \; r_2(x, y) \geq b$ for some $b \in \mathbb{R}$. Now, given the four datasets and the constrained optimization approach, we wish to compare learning the optimal policy as described by a Pareto frontier in the $(r_1, r_2)$ space. We train a neural network policy with DPO (Rafailov et al., 2023) for each of the four datasets, and solve a constrained optimization problem for the COP approach. See Figure 9(a) for empirical results under reward model sets $A$ (left) and $B$ (right).

While it is somewhat trivial to see why learning from $\mathcal{D}_1$ and $\mathcal{D}_2$ alone yields suboptimal rewards, the case for the jointly preferred dataset $\mathcal{D}_J$ and a convex weighted reward model dataset $\mathcal{D}_C$ is not obvious. The issue with $\mathcal{D}_J$ is that it only contains samples where $r_1$ and $r_2$ agree, effectively discarding all points that exhibit a meaningful trade-off between the two objectives. This results in a sparse and biased preference signal that does not span the entire Pareto front. In contrast, $\mathcal{D}_C$ encodes preferences with respect to a *single* scalarized reward model, which inherently biases learning toward one specific convex combination of the objectives. While more sophisticated approaches have been proposed for learning from multi-dimensional preferences - such as RiC (Yang et al., 2024), MODPO (Zhou et al., 2023), Rewarded Soups (Rame et al., 2023), and SIPO (Li et al., 2025) - they remain fundamentally limited in their expressivity. Ultimately, each method relies on learning with respect to a *single* scalarized reward signal of the form $\widetilde{r} = f(r_1, r_2)$, where the function $f$ varies across methods. As a result, these approaches do not recover the full structure of the underlying preference landscape and cannot characterize the Pareto front in the multi-objective setting.

In contrast, constrained optimization (COP) over $r_1$ and $r_2$ yields solutions that lie close to the true Pareto frontier. This highlights the need for optimization methods that explicitly account for trade-offs across objectives, rather than collapsing them into a single reward signal, in order to fully leverage multi-dimensional preference data. MOPO follows this principle by directly optimizing within the multi-objective space, and empirically achieves solutions that approach the Pareto front as in Figure 9(b).

We further empirically validate the correctness of MOPO and consider whether it is able to generalize and regularize effectively w.r.t. the reference policy. See Figure 10 for comparison of the policy learned through MOPO under various regularization values. We observe that even with an uninformed $\pi_{\text{ref}}$, MOPO is able to push toward the Pareto frontier and is limited only by the strength of regularization.

**Remark A.1.1.** *Note that reward models are only used for evaluation, and are not assumptions or requirements to finetune policies using MOPO. MOPO learns strictly from preference data, without: (i) assuming the existence of a mapping from preferences to pointwise rewards, and (ii) learning this mapping (reward model) from preference data.*

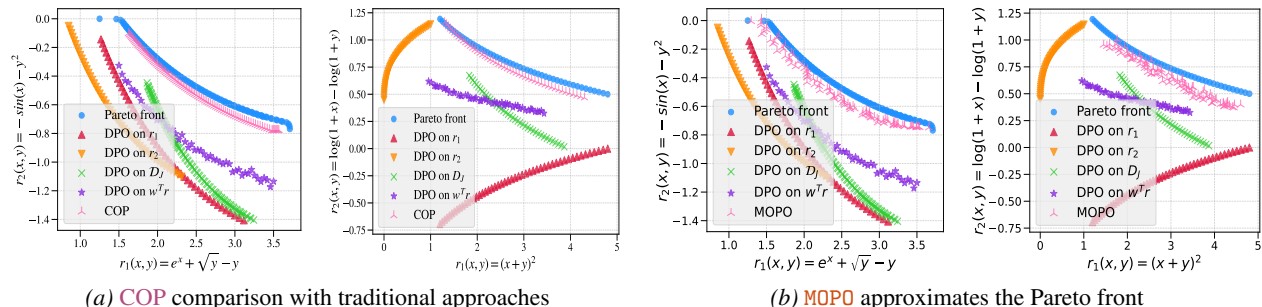

*(a)* COP *comparison with traditional approaches*      *(b)* MOPO *approximates the Pareto front*

*Figure 9.* Illustration of how a COP approach, and hence MOPO, achieves Pareto-optimal alignment in comparison with DPO on $\mathcal{D}_1$, $\mathcal{D}_2$, $\mathcal{D}_J$, and $\mathcal{D}_C$ under two sets of reward models.

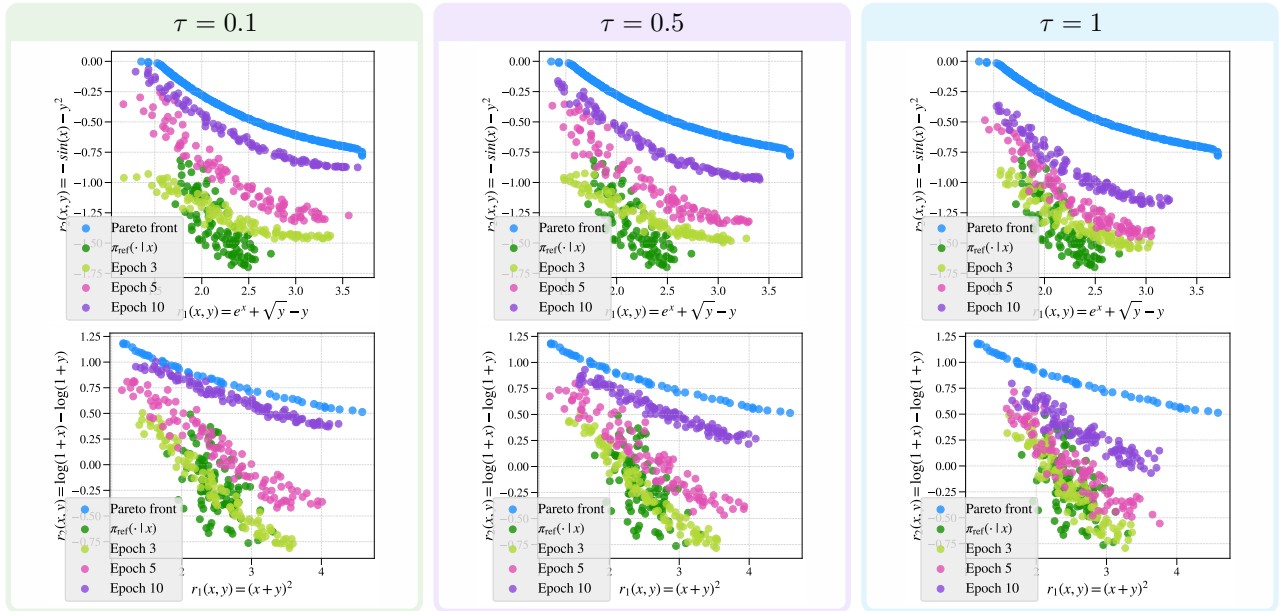

*Figure 10.* Comparison of the KL-regularized policy learned using MOPO with the reference policy $\pi_{\text{ref}}$ and the Pareto frontier, visualized in the reward space for two reward model pairs, $A$ (top) and $B$ (bottom).

## A.2. Sub-optimality of baselines.

To begin, we list some common divergence metrics that have been used in literature to characterize two probability distributions. Then, we follow with some definitions before listing the main proofs for the sub-optimality of baselines.

### A.2.1. DIVERGENCE MEASURES AND CLOSED-FORM POLICIES

We acknowledge that commonly used $f$-divergence measures have been introduced in (Wang et al., 2024a; Shi et al., 2024) and show them here for completeness:

| Divergence measure | $f(x)$ | $\nabla f(x)$ | barrier function |
|---|---|---|---|
| Reverse KL-divergence | $x \log x$ | $\log x + 1$ | ✔ |
| Forward KL-divergence | $-\log x$ | $-1/x$ | ✔ |
| JSD | $x \log x - (x+1) \log \frac{x+1}{2}$ | $\log \frac{2x}{1+x}$ | ✔ |
| $\alpha$-divergence | $\frac{x^{1-\alpha} - (1-\alpha)x - \alpha}{\alpha(1-\alpha)}$ | $(1 - x^{-\alpha})/\alpha$ | ✔ |
| Jeffery divergence | $x \log x - \log x$ | $\log x - \frac{1}{x} + 1$ | ✔ |
| Total Variation | $|x-1|/2$ | $\mathrm{sgn}(x-1)/2$ | ✗ |
| Chi-squared | $(x-1)^2$ | $2(x-1)$ | ✗ |

Here we show the optimal sampling policies for multi-objective w.r.t. these divergence measures:

| Divergence measure | Optimal policy |
|---|---|
| Reverse KL-divergence | $\left( \prod_{i=1}^{K} \pi_i(y|x)^{w_i} \right) \cdot \exp(-Z(x))$ |
| Forward KL-divergence | $\pi_{\mathsf{ref}}(y|x) \cdot \left( Z(x) + \sum_{i=1}^{K} \frac{w_i \pi_{\mathsf{ref}}(y|x)}{\pi_i(y|x)} \right)^{-1}$ |
| JSD | $\pi_{\mathsf{ref}}(y|x) \cdot \left( -1 + \exp(Z(x)) \prod_{i=1}^{K} \left( \frac{\pi_{\mathsf{ref}}(y|x)}{\pi_i(y|x)} + 1 \right)^{w_i} \right)^{-1}$ |
| $\alpha$-divergence | $\pi_{\mathsf{ref}}(y|x) \cdot \left( \alpha Z(x) + \sum_{i=1}^{K} w_i \left( \frac{\pi_{\mathsf{ref}}(y|x)}{\pi_i(y|x)} \right)^{\alpha} \right)^{-\frac{1}{\alpha}}$ |

### A.2.2. DEFINITIONS

We first begin with some definitions.

**Definition A.2.1** ($f$-divergence (Ali & Silvey, 1966; Agnihotri et al., 2023; 2024c))**.** *For probability measures $P$ and $Q$, let $\mu$ be a dominating measure of $P$ and $Q$ (i.e. $P, Q \ll \mu$), and let $p, q$ be the Radon-Nikodym derivative (Durrett, 2010) $\frac{dP}{d\mu}$, $\frac{dQ}{d\mu}$ respectively. For simplicity, here we assume $q > 0$ almost surely. Then $f$-divergence from $P$ to $Q$ is defined as*

$$I_f(p\|q) := \int q f\left( \frac{p}{q} \right) d\mu \,,$$

*where $f$ is convex on $\mathbb{R}_+$, satisfying $f(1) = 0$. Most useful divergence measures are included in $f$-divergences, and the commonly used ones and corresponding $f$ are introduced in Appendix A.2.1.*

**Definition A.2.2** (Barrier function (Nesterov, 2018; Agnihotri et al., 2026))**.** *Given conditions satisfied in Definition A.2.1, if additionally $0 \notin \mathrm{dom}(\nabla f)$, then $f$ is a barrier function. If a barrier function $f$ is continuously differentiable and strongly convex on $\mathbb{R}_+$, then $f$ is a strongly convex and smooth barrier function (abbreviated as strong-barrier function).*

**Definition A.2.3** (Expected calibration error (Guo et al., 2017; Wang et al., 2024a))**.** *Denote the ground truth distribution as $\mathbb{P}$, context as $X$ and response as $Y$. The expected calibration error of a stochastic policy $\pi$ is defined as*

$$\mathrm{ECE}(\pi) := \mathbb{E}_{\substack{x \sim \mathcal{X} \\ y \sim \pi(\cdot|x)}} \left| \mathbb{P}(Y = y|X = x) - \pi(y|x) \right| \,.$$

**Hypothesis 1** (Reducible reward misspecification (Wortsman et al., 2022; Rame et al., 2023; Jang et al., 2023))**.** *Let $\theta_k$ be the parameter of the optimal policy for objective value $J_k$, $\forall k \in [K]$, and $\theta_w^\star$ be the parameter of the optimal policy for the interpolated objective $\sum_{k=1}^{K} w_k \cdot J_k$, then this hypothesis claims that*

$$\theta_w^* \in \left\{ \sum_{k=1}^{K} \lambda_k \cdot \theta_k, \lambda \in \Delta^{K-1} \right\} \,, \ \forall w \in \Delta^{K-1} \,.$$

Extending the results of (Wang et al., 2024a) to the multi-objective setting, we prove the necessity of $f$ being barrier functions to find an optimal policy $\pi^\star$ for multi-objective alignment. We refer the reader to (Shi et al., 2024) for a complete discussion.

**Theorem A.2.4** (Necessity of barrier functions (Shi et al., 2024)). *If $f$ is not a barrier function, then for $\forall C \in \mathbb{R}_+$, $N \in \mathbb{Z}_{\geq 4}$, $K \in \mathbb{Z}_{\geq 2}$, $\mathcal{Y} = \{y_i\}_{i=1}^N$, any multi-objective decoding or merging algorithm $\mathcal{A} : \mathcal{S}^{K+1} \times \Delta^{K-1} \to \mathcal{S}$, there exists a reference policy $\pi_{\mathsf{ref}}$, policies $\{\pi_i\}_{i=1}^K$ and $\pi'$, reward functions $\{\mathcal{R}_i\}_{i=1}^K$, preference weightings $w \in \Delta^{K-1}$ and $\tau \in \mathbb{R}_+$, s.t. $\pi_i$ is the optimal policy for $\mathcal{R}_i$ w.r.t. $\tau \cdot I_f(\cdot \| \pi_{\mathsf{ref}})$, $\forall i \in [K]$, but*

$$\mathbb{E}_{y \sim \pi_{\mathcal{A},w}} \left[ \sum_{i=1}^K w_i \mathcal{R}_i(y) \right] \leq \mathbb{E}_{y \sim \pi'} \left[ \sum_{i=1}^K w_i \mathcal{R}_i(y) \right] - C \text{ , and}$$

$$\mathbb{E}_{y \sim \pi_{\mathcal{A},w}} \left[ \sum_{i=1}^K w_i \mathcal{R}_i(y) \right] - \tau I_f(\pi_{\mathcal{A},w} \| \pi_{\mathsf{ref}}) \leq \mathbb{E}_{y \sim \pi'} \left[ \sum_{i=1}^K w_i \mathcal{R}_i(y) \right] - \tau I_f(\pi' \| \pi_{\mathsf{ref}}) - C \text{ ,}$$

*where $\pi_{\mathcal{A},w}(y) := \mathcal{A}\big(\pi_{\mathsf{ref}}, \pi_1, \pi_2, \ldots, \pi_K, w\big)(y)$ .*

**Remark A.2.5** (Motivating example). *Here we provide a motivating example where $f \equiv 0$: let $K = 4$, $\mathcal{R}_1(y_1) = \mathcal{R}_2(y_2) = 1$, $\mathcal{R}_1(y_2) = \mathcal{R}_2(y_1) = -1$, $\mathcal{R}_1(y_{3+k}) = \mathcal{R}_2(y_{3+k}) = 0$, $\mathcal{R}_1(y_{4-k}) = \mathcal{R}_2(y_{4-k}) = 1/2$, where $k \in \{0, 1\}$. Then the optimal policy for $\mathcal{R}_1$ is $\pi_1(y_i) := \delta_{1i}$, for $\mathcal{R}_2$ is $\pi_2(y_i) := \delta_{2i}$, and for $\mathcal{R}_1/2 + \mathcal{R}_2/2$ is $\pi^\star(y_i) := \delta_{4-k,i}$. Thus $\pi_{\mathcal{A},w}$ cannot fit $\pi^\star$ both for $k = 0, 1$.*

*Proof.* Since $f$ is not a barrier function, $0 \in \mathrm{dom}(\nabla f)$. Now we can define $p := \max_{x \in [0,N]} \nabla f(x)$, $q := \min_{x \in [0,N]} \nabla f(x)$, $r := \max_{x \in [0,N]} f(x) - \min_{x \in [0,N]} f(x)$, $s := \frac{N-2}{N-3} \cdot C$. Let $w = (0.5, 0.5, \underbrace{0, \ldots, 0}_{N-2})$, and we pick $k = \underset{j \in \{3,4,\ldots,N\}}{\mathrm{argmin}} \pi_{\mathcal{A},w}(y_j)$.

Let $\pi_{\mathsf{ref}}(y_i) = \frac{1}{N}$, $\pi_1(y_i) = \delta_{1i}$, $\pi_2(y_i) = \delta_{2i}$, $\pi_j(y_i) = \frac{1}{N}$ and $\pi'(y_i) = \delta_{ik}$, $\forall i \in [N]$, $j \in \{3, 4, \ldots, K\}$. And set

$$\mathcal{R}_1(y_i) = \begin{cases} 2p + 2r + 2s & i = 1 \\ 4q - 2p - 2r - 2s & i = 2 \\ p + q + r + s & i = k \\ 2q & \text{o/w} \end{cases}, \mathcal{R}_2(y_i) = \begin{cases} 4q - 2p - 2r - 2s & i = 1 \\ 2p + 2r + 2s & i = 2 \\ p + q + r + s & i = k \\ 2q & \text{o/w} \end{cases}, \text{and } \mathcal{R}_j \equiv 0, \forall j \in \{3, 4, \ldots, K\}.$$

Let $\tau = 1$, then the optimization objective for $\mathcal{R}_1$ w.r.t. $I_f$ is $J_1(\pi) := \mathbb{E}_{y \sim \pi} [\mathcal{R}_1(y)] - I_f(\pi \| \pi_{\mathsf{ref}})$, and the Lagrangian dual is

$$\mathcal{L}_1(\pi) := \sum_{i=1}^N \left( -\mathcal{R}_1(y_i) \cdot \pi(y_i) + \frac{1}{N} f\left(N \cdot \pi(y_i)\right) \right) + \lambda \left( \sum_{i=1}^N \pi(y_i) - 1 \right) - \sum_{i=1}^N \mu_i \pi(y_i) \text{ .}$$

As the objective is convex and the constraints are affine, we can directly apply the *Karush-Kuhn-Tucker conditions* (Nesterov, 2018):

$$\nabla \mathcal{L}_1(\pi_1^\star) = 0 \text{ ,} \tag{12}$$

$$\sum_{i=1}^N \pi_1^\star(y_i) = 1 \text{ ,}$$

$$\pi_1^\star(y_i) \geq 0 \text{ ,}$$

$$\mu_i^\star \geq 0 \text{ ,}$$

$$\mu_i^\star \pi_1^\star(y_i) = 0 \text{ .} \tag{13}$$

Equation (12) implies

$$-\mathcal{R}_1(y_i) + \nabla f(N \cdot \pi_1^\star(y_i)) + \lambda^\star - \mu_i^\star = 0 \text{ .}$$

If $\pi_1^\star(y_1) > 0$, we have

$$\lambda^\star = \mathcal{R}_1(y_1) - \nabla f(N \cdot \pi_1^\star(y_1))$$
$$\geq p + 2r + 2s \text{ ,}$$

and then for $\forall j \neq 1$ ,

$$
\begin{aligned}
\mu_j^\star &= -\mathcal{R}_1(y_j) + \nabla f(N \cdot \pi_1^\star(y_j)) + \lambda^\star \\
&\geq -p - q - r - s + q + p + 2r + 2s \\
&= r + s \\
&> 0 .
\end{aligned}
$$

Combining it with Equation (13) yields $\pi_1^\star(y_j) = 0$ for $\forall j \neq 1$, which is exactly $\pi_1$. Note that we have

$$
J(\pi_1) \geq 2p + 2r + 2s - \max_{x \in [0,N]} f(x) .
$$

For any $\pi'$ with $\pi'(y_1) = 0$, we have

$$
\begin{aligned}
J(\pi') &\leq p + q + r + s - \min_{x \in [0,N]} f(x) \\
&= p + q + 2r + s - \max_{x \in [0,N]} f(x) \\
&< J(\pi_1) .
\end{aligned}
$$

Thus $\pi_1$ is the optimal policy for $\mathcal{R}_1$ w.r.t. $I_f(\cdot \| \pi_{\mathsf{ref}})$. Similarly, $\pi_2$ is the optimal policy for $\mathcal{R}_2$ w.r.t. $I_f(\cdot | \pi_{\mathsf{ref}})$. By convexity of $f$, the minimum of $I_f(\pi \| \pi_{\mathsf{ref}})$ is obtained when $\pi = \pi_{\mathsf{ref}}$, and thus $\pi_j$ is the optimal policy for $\mathcal{R}_j$ w.r.t. $I_f(\cdot \| \pi_{\mathsf{ref}})$, for $\forall j \in \{3, 4, \ldots, K\}$. Therefore, all conditions are well satisfied by this construction. Note that

$$
\mathbb{E}_{y \sim \pi'} \left[ \sum_{i=1}^K w_i \mathcal{R}_i(y) \right] = p + q + r + s . \tag{14}
$$

While by the selection of $k$, we have

$$
\mathbb{E}_{y \sim \pi_{\mathcal{A},w}} \left[ \sum_{i=1}^K w_i \mathcal{R}_i(y) \right] \leq \frac{(N-3) \cdot 2q + p + q + r + s}{N-2} . \tag{15}
$$

Comparing Equation (14) with Equation (15), we have

$$
\begin{aligned}
\mathbb{E}_{y \sim \pi_{\mathcal{A},w}} \left[ \sum_{i=1}^K w_i \mathcal{R}_i(y) \right] &\leq \mathbb{E}_{y \sim \pi'} \left[ \sum_{i=1}^K w_i \mathcal{R}_i(y) \right] - \frac{N-3}{N-2} s \\
&= \mathbb{E}_{y \sim \pi'} \left[ \sum_{i=1}^K w_i \mathcal{R}_i(y) \right] - C .
\end{aligned}
$$

Note that $\pi_{\mathsf{ref}}$ is a uniform distribution and both $\pi_{\mathcal{A},w}, \pi'$ are one-point distributions, thus $I_f(\pi_{\mathcal{A},w} \| \pi_{\mathsf{ref}}) = I_f(\pi' \| \pi_{\mathsf{ref}})$. We have

$$
\mathbb{E}_{y \sim \pi_{\mathcal{A},w}} \left[ \sum_{i=1}^K w_i \mathcal{R}_i(y) \right] - I_f(\pi_{\mathcal{A},w} \| \pi_{\mathsf{ref}}) \leq \mathbb{E}_{y \sim \pi'} \left[ \sum_{i=1}^K w_i \mathcal{R}_i(y) \right] - I_f(\pi' \| \pi_{\mathsf{ref}}) - C . \qquad \square
$$

### A.2.3. BASELINES ARE NOT PARETO-OPTIMAL

Given the necessity of $f$ being a barrier-function, we now show how parameter-merging paradigm algorithms ((Guo et al., 2024; Yang et al., 2024; Jang et al., 2023; Zhou et al., 2023)) fail to achieve Pareto-optimality. The optimality of parameter-merging paradigm primarily relies on reduced reward mis-specification hypothesis (see Hypothesis 1). The following theorem from Shi et al. (2024) demonstrates that this hypothesis does not hold for almost all $f$-divergence regularized policies.

**Theorem A.2.6.** *For any $f$-divergence satisfying one of the following conditions: (i) $f$ is not a barrier function; (ii) $I_f$ is Reverse KL-divergence; (iii) $f$ is a strong-barrier function, with finite roots of*

$$2\nabla f\left(\frac{3\sqrt{1-2x}}{2\sqrt{1-2x}+\sqrt{x}}\right) - 2\nabla f\left(\frac{3\sqrt{x}}{2\sqrt{1-2x}+\sqrt{x}}\right) - \nabla f(3-6x) + \nabla f(3x) ,$$

*$\exists N, K \in \mathbb{N}$, $\mathcal{Y} = \{y_i\}_{i=1}^N$, $\tau \in \mathbb{R}_+$, a neural network $nn = \mathrm{softmax}(h_\theta(z_0))$ where $z_0 \in \mathbb{R}^n$ and $h_\theta : \mathbb{R}^n \to \mathbb{R}^N$ is a continuous mapping, preference weightings $w \in \Delta^{K-1}$, reference policy $\pi_{\mathsf{ref}}$, and the objectives $J_1, J_2, \ldots, J_K$ representing reward functions $\mathcal{R}_1, \mathcal{R}_2, \ldots, \mathcal{R}_K$ w.r.t. $\tau \cdot I_f(\cdot \| \pi_{\mathsf{ref}})$, s.t. Hypothesis 1 does not hold.*

*Proof.* (i) If $f$ is not a barrier function, Hypothesis 1 does not hold immediately from Theorem A.2.4.

(ii) If $I_f$ is Reverse KL-divergence, we let $N = 3$, $K = 3$, and $h_\theta(z_0) = W_\theta^{(2)} \sigma\left(W_\theta^{(1)} z_0\right)$, where $\sigma$ is ReLU$(\cdot)$. We set $\mathcal{R}_i(y_j) = \delta_{ij}$, $\pi_{\mathsf{ref}}(y_i) = 1/3$ for $\forall i, j \in [3]$, $z_0 = 1$ and $\tau = 1$. Then the optimal policies are $W_{\theta_1}^{(1)} = e_1$, $W_{\theta_1}^{(2)} = \begin{pmatrix} 100 \\ 000 \\ 000 \end{pmatrix}$ for $\mathcal{R}_1$ w.r.t. KL·$\pi_{\mathsf{ref}}$, $W_{\theta_2}^{(1)} = e_2$, $W_{\theta_2}^{(2)} = \begin{pmatrix} 000 \\ 010 \\ 000 \end{pmatrix}$ for $\mathcal{R}_2$ w.r.t. KL·$\pi_{\mathsf{ref}}$, and $W_{\theta_3}^{(1)} = e_3$, $W_{\theta_3}^{(2)} = \begin{pmatrix} 000 \\ 000 \\ 001 \end{pmatrix}$ for $\mathcal{R}_3$ w.r.t. KL·$\pi_{\mathsf{ref}}$. Thus we have $h_{\sum_{j=1}^3 \lambda_j \theta_j}(z_0) = \left(\lambda_1^2, \lambda_2^2, \lambda_3^2\right)^\top$. Given $w = (0, 1/3, 2/3)$, the optimal policy $\pi^\star$ should output $\pi^\star(y_1) = \frac{1}{1+\exp(1/3)+\exp(2/3)}$, $\pi^\star(y_2) = \frac{\exp(1/3)}{1+\exp(1/3)+\exp(2/3)}$ and $\pi^\star(y_3) = \frac{\exp(2/3)}{1+\exp(1/3)+\exp(2/3)}$. Note that

$$\sqrt{t} + \sqrt{t+1/3} + \sqrt{t+2/3} > 1 , \ \forall t \in \mathbb{R}_+ ,$$

thus there is no solution $\lambda \in \Delta^2, t \in \mathbb{R}_+$ for $\left(\lambda_1^2, \lambda_2^2, \lambda_3^2\right)^\top = \left(t, t + \frac{1}{3}, t + \frac{2}{3}\right)^\top$, *i.e.* there is no $\lambda$ s.t. $\mathrm{softmax}\left(h_{\sum_{j=1}^3 \lambda_j \theta_j}(z_0)\right) = (\pi^\star(y_1), \pi^\star(y_2), \pi^\star(y_3))$, *i.e.* Hypothesis 1 does not hold.

(iii) If $f$ is a strong-barrier function, with finite roots of

$$2\nabla f\left(\frac{3\sqrt{1-2x}}{2\sqrt{1-2x}+\sqrt{x}}\right) - 2\nabla f\left(\frac{3\sqrt{x}}{2\sqrt{1-2x}+\sqrt{x}}\right) - \nabla f(3-6x) + \nabla f(3x) ,$$

we let $N = 3$, $K = 2$, $h_\theta(z_0) = W_\theta(z_0)$, $z_0 = 1$, $\mathcal{R}_1(y_i) = \delta_{1i}$, $\mathcal{R}_2(y_i) = \delta_{2i}$ and $\pi_{\mathsf{ref}}(y_i) = 1/3$, for $\forall i \in [3]$. The optimal policy for $J_1$ is $\pi_{\theta_1}(y_i) = \frac{1}{3}(\nabla f)^{(-1)}\left(\frac{1}{\tau}\delta_{1i} - Z\right)$, and the optimal policy for $J_2$ is $\pi_{\theta_2}(y_i) = \frac{1}{3}(\nabla f)^{(-1)}\left(\frac{1}{\tau}\delta_{2i} - Z\right)$, where $Z$ is the normalization factor. And these policies can be learned by setting $W_{\theta_i} = \left(\log \pi_{\theta_i}(y_1), \log \pi_{\theta_i}(y_2), \log \pi_{\theta_i}(y_3)\right)^\top$.

We set $a := \pi_{\theta_1}(y_1) = \frac{1}{3}(\nabla f)^{(-1)}(\frac{1}{\tau} - Z)$, $b := \pi_{\theta_1}(y_2) = \pi_{\theta_1}(y_3) = \frac{1}{3}(\nabla f)^{(-1)}(-Z)$. Thus we have

$$\nabla f(3a) - \nabla f(3b) = \frac{1}{\tau} , \tag{16}$$

$$a + 2b = 1 . \tag{17}$$

The optimal policy for $w_1 \cdot J_1 + w_2 \cdot J_2$ (see (Shi et al., 2024) for proof) is

$$\pi_w^\star(y_i) = \frac{1}{3}(\nabla f)^{(-1)}\left(-Z_w^\star + \frac{w_1}{\tau}\delta_{1i} + \frac{w_2}{\tau}\delta_{2i}\right) , \tag{18}$$

where $Z_w^\star$ is the normalization factor. By linearly merging the weights of $\pi_{\theta_1}$ and $\pi_{\theta_2}$, we have

$$\pi_{\lambda_1 \theta_1 + \lambda_2 \theta_2}(y_i) = \mathrm{softmax}\left(\lambda_1 W_{\theta_1}(z_0) + \lambda_2 W_{\theta_2}(z_0)\right)(y_i)$$

$$= \frac{1}{Z_\lambda}\left((\nabla f)^{(-1)}\left(\frac{1}{\tau}\delta_{1i} - Z\right)\right)^{\lambda_1}\left((\nabla f)^{(-1)}\left(\frac{1}{\tau}\delta_{2i} - Z\right)\right)^{\lambda_2} , \tag{19}$$

where $Z_\lambda$ is the normalization factor.

With symmetry, Equation (18), (19) and Hypothesis 1 indicate that $\pi_{\frac{1}{2}\theta_1 + \frac{1}{2}\theta_2} = \pi^\star_{(\frac{1}{2}, \frac{1}{2})}$, thus

$$\frac{1}{3}(\nabla f)^{(-1)} \left( -Z^\star_{(0.5, 0.5)} + \frac{1}{2\tau} \right) = \frac{\sqrt{a}}{2\sqrt{a} + \sqrt{b}},$$

$$\frac{1}{3}(\nabla f)^{(-1)} \left( -Z^\star_{(0.5, 0.5)} \right) = \frac{\sqrt{b}}{2\sqrt{a} + \sqrt{b}},$$

and combining them with Equation (16) yields

$$2\nabla f \left( \frac{3\sqrt{a}}{2\sqrt{a} + \sqrt{b}} \right) - 2\nabla f \left( \frac{3\sqrt{b}}{2\sqrt{a} + \sqrt{b}} \right) = \nabla f(3a) - \nabla f(3b). \tag{20}$$

Given the condition, the solution set $(a, b)$ to Equation (17), (20) is finite, thus there exists $\tau \in \mathbb{R}_+$ s.t. Equation (16) does not hold, implying that Hypothesis 1 does not hold. $\qquad\square$

### A.3. Constraint Threshold Initialization

We begin this section by describing an equivalence between the preference-learning based optimization problem and a reward-learning based optimization problem. Note that `MOPO` does not assume this equivalence, and directly works with preference data, and this equivalence is established for analysis only. Following this equivalence discussion, we specify a provable method for setting constraint thresholds $b$ such that the optimal solution learned by `MOPO` is a Pareto-optimal solution.

For all preference instances in the preference dataset, there exists an underlying, unknown reward model based on which preferences are provided. For all contexts-output pairs $(x, y) \in \mathcal{X} \times \mathcal{Y}$, let the reward model for the $k^{th}$ objective be $r_k(x, y) \in \mathbb{R}$ for $k \in [K]$.

**Preference–reward link.** For every objective $k \in [K]$ there exists a *strictly increasing* function $\phi_k : \mathbb{R} \to \mathbb{R}$ and a strictly increasing transfer function $\sigma : \mathbb{R} \to (0, 1)$ such that for all contexts $x \in \mathcal{X}$ and actions $y, y' \in \mathcal{Y}$ we have,

$$p_k(y \succ y' \mid x) = \sigma\big(\phi_k\big(r_k(x, y) - r_k(x, y')\big)\big). \tag{21}$$

Now, with respect to the $k^{th}$ objective, let the expected reward $R_k(\pi)$ and preference-based objective value $F_k(\pi)$ for a policy be given by,

$$R_k(\pi) := \mathbb{E}_{x \sim \nu, \, y \sim \pi(\cdot|x)}[r_k(x, y)] \text{ for } k \in [K],$$

$$F_k(\pi) := \mathbb{E}_{x \sim \nu, \, y \sim \pi(\cdot|x), \, y' \sim \mu(\cdot|x)}\big[p_k(y \succ y' \mid x)\big], \quad k \in [K-1],$$

$$F_K(\pi) := \mathbb{E}_{x \sim \nu, \, y \sim \pi(\cdot|x), \, y' \sim \mu(\cdot|x)}\big[p(y \succ y' \mid x)\big] - \tau \, \mathsf{KL}\big(\pi \parallel \pi^{\text{ref}}\big).$$

**Lemma A.3.1** (Order preservation). *Under Equation (21) and some $u \in \mathbb{R}$, the mapping $H_k(u) := \mathbb{E}_{z \sim \text{Unif}[-u, u]}\big[\sigma \circ \phi_k(z)\big]$ is strictly increasing. Moreover, for every policy $\pi$*

$$F_k(\pi) = H_k\big(R_k(\pi) - R_k(\mu)\big), \qquad k \in [K-1],$$

*so that for any $\pi, \pi'$, we have $R_k(\pi) \geq R_k(\pi') \iff F_k(\pi) \geq F_k(\pi')$.*

*Proof.* Fix $k$. By Equation (21), $\sigma \circ \phi_k$ is strictly increasing, hence so is its odd extension $z \mapsto \sigma \circ \phi_k(z)$. For $U := R_k(\pi) - R_k(\mu)$ let $z := r_k(x, y) - r_k(x, y')$. Because $(y, y') \sim (\pi, \mu)$ are independent, $z$ is symmetrically distributed around $U$ and $z \sim \text{Unif}[U - \delta, U + \delta]$ for some $\delta > 0$ that does not depend on $U$. Taking expectation yields $F_k(\pi) = H_k(U)$, and strict monotonicity of $H_k$ follows from strict monotonicity of $\sigma \circ \phi_k$. $\qquad\square$

Note there that we introduced $H_k(u) = \mathbb{E}_{z\sim\text{Unif}[-u,u]}\big[\sigma \circ \phi_k(z)\big]$. The uniform law is chosen purely for notational brevity; the proof requires only that the base distribution be symmetric and shifted by the reward gap $u = R_k(\pi) - R_k(\mu)$. Consequently, one may replace $\text{Unif}[-u,u]$ by any symmetric density $\rho_u(z) = \rho(z-u)$, and define $H_k(u) = \mathbb{E}_{z\sim\rho_u}\big[\sigma \circ \phi_k(z)\big]$. Strict monotonicity of $\sigma \circ \phi_k$ then guarantees that the Lemma A.3.1 holds. Now, let $\boldsymbol{b} \in [0,1]^{K-1}$ be the probability thresholds in COP, let $\mathbf{F}_{1:K-1}(\pi) = (F_1(\pi), \ldots, F_{K-1}(\pi))$ and $\mathbf{R}_{1:K-1}(\pi) = (R_1(\pi), \ldots, R_{K-1}(\pi))$, and define component-wise

$$c_k := H_k^{-1}(b_k) + R_k(\mu), \quad k \in [K-1], \quad \boldsymbol{c} := (c_1, \ldots, c_{K-1}).$$

By Lemma A.3.1, $F_k(\pi) \geq b_k \iff R_k(\pi) \geq c_k$. Hence the preference-space constrained problem

$$\max_\pi \ F_K(\pi) \ \texttt{s.t.} \ \mathbf{F}_{1:K-1}(\pi) \geq \boldsymbol{b} \tag{COP}$$

is *equivalent* to the reward-space problem

$$\max_\pi \big\{ R_K(\pi) - \tau\,\text{KL}(\pi \parallel \pi^{\text{ref}}) \big\} \ \texttt{s.t.} \ \mathbf{R}_{1:K-1}(\pi) \geq \boldsymbol{c}. \tag{COP-R}$$

Given this equivalence, we now describe the procedure of setting appropriate constraint thresholds $\boldsymbol{c}$ for the COP-R problem, following which constraint thresholds $\boldsymbol{b}$ for the original COP can be obtained element-wise via $c_k := H_k^{-1}(b_k) + R_k(\mu)$. This setting of constraint thresholds ensures that the optimal solution of the COP problem (Problem (1)) is also a Pareto-optimal solution. We begin with a definition, following which we state the main result for $\boldsymbol{c}_k$, which holds for all $k \in [K-1]$.

**Definition A.3.2** (Insertion Index)**.** *Let* $\mathsf{P}_k = \big(\mathsf{P}_k(0), \mathsf{P}_k(1), \ldots, \mathsf{P}_k(M-1)\big)$ *be an ascending (sorted) list of the $k$-th objective values from the Pareto front consisting of $M$ points. For any new value $\alpha \in \mathbb{R}$, the insertion index $j$ is the smallest integer $0 \leq j < M$ satisfying $\mathsf{P}_k(j) \geq \alpha$ (if such a $j$ exists), and set $j = M$ if no such index exists.*

**Proposition A.3.3.** *For the initial point $\pi_0$ of Problem (1), let the insertion index of $\alpha_k := R_k(\pi_0)$ in $\mathsf{P}_k$ be $j_k$. If $\boldsymbol{c}_k \geq \mathsf{P}_k(\max(0, j_k-1)) \ \forall\, k \in [K-1]$, then the optimal solution of Problem COP-R, if it exists, is a Pareto-optimal solution.*

*Proof.* We prove by contradiction. First, define a solution element by the tuple $(\pi, \mathbf{P}^\pi)$, which refers to a policy $\pi$ along with its corresponding reward vector $\mathbf{P}^\pi = (R_1(\pi), \ldots, R_K(\pi))$. Now suppose that the optimal solution $P' = (\pi', \mathbf{P}^{\pi'})$ of Problem COP-R is not a Pareto-optimal solution. By the definition of Pareto-optimal solution, there exists a solution $\widehat{P} = (\widehat{\pi}, \mathbf{P}^{\widehat{\pi}})$ in $\Pi_P$ that dominates $P'$, i.e., $R_k(\pi') \leq R_k(\widehat{\pi}) \ \forall\, k \in [K]$. Given $P_0 = (\pi_0, \mathbf{P}^{\pi_0})$, we have $R_K(\widehat{\pi}) \geq R_K(\pi') \geq R_K(\pi_0)$ by definition. Since both $P_0$ and $\widehat{P}$ do not dominate each other, since $R_K(\widehat{\pi}) \geq R_K(\pi_0)$, there exists $k \in [K-1]$ such that $R_k(\pi_0) \geq R_k(\widehat{\pi})$.

Now consider the values of $\boldsymbol{c}_k$ and $R_k(\widehat{\pi})$ for some objective $k$. Note that $\boldsymbol{c}_k \geq \mathsf{P}_k(\max(0, j_k-1))$. If $R_k(\pi_0) \geq R_k(\widehat{\pi}) > \boldsymbol{c}_k$, then $R_k(\pi_0) \geq R_k(\widehat{\pi}) > \mathsf{P}_k(\max(0, j_k-1))$, which is conflicting with the condition that $\mathsf{P}_k(\max(0, j_k-1))$ is the $(\max(0, j_k-1))^{th}$ objective value in $\mathsf{P}_k$. If $R_k(\widehat{\pi}) \leq \boldsymbol{c}_k$, it conflicts with the condition that $\widehat{P}$ dominates $P'$. Therefore, such a $\widehat{P}$ does not exist, and hence, $P'$ is a Pareto-optimal solution. $\square$

Proposition A.3.3 formalizes the criteria for specifying appropriate constraint values and provides the condition for which the optimal solution of Problem (1) is a Pareto optimal solution. See Figure 11 for the visualization. Proposition A.3.3 gives a *sufficient* condition under which the solution of Problem (2) is Pareto optimal. However, in practice this condition is (i) overly conservative and may exclude many feasible Pareto points, and (ii) computationally expensive as it requires re-evaluating *all* policies for non-dominated sorting at every optimization step. Please see Section 3.1 for an empirically validated practical constraint specification procedure.

### A.4. Log-barrier Function Optimization

For some $\sigma, s > 0$, consider the following log barrier function,

$$\mathcal{B}_{\sigma,s}(z) = \begin{cases} -\sigma \log(-z), & z \leq -s \\ \frac{\sigma}{s}z + (1-\log(s))\sigma, & z > -s \end{cases}, \quad \text{with } \partial_z \mathcal{B}_{\sigma,s}(z) = \frac{\sigma}{\max(-z,s)}.$$

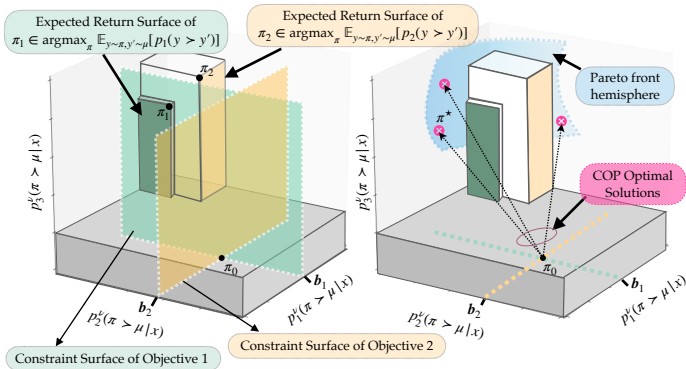

*Figure 11.* Visualization of criteria for specifying constraint values. The expected return surface of $\pi_1$ ($\pi_2$), call it $\mathcal{S}_1$ ($\mathcal{S}_2$), in objective 1(2) is the $\max(0, j_k - 1)^{th}$ value in list $\mathsf{P}_1$ ($\mathsf{P}_2$) respectively. Therefore, specifying constraints values $\boldsymbol{b}_1 \geq \mathcal{S}_1$ and $\boldsymbol{b}_2 \geq \mathcal{S}_2$ is sufficient for the optimal solution of Equation (1) to be a Pareto-optimal solution.

For all $z \in \mathbb{R}$, this is a convex, continuous, and differentiable function. Importantly, for $s = \sigma^2$, this barrier function converges to the characteristic function $\chi\{z \leq 0\}$ as $\sigma \to 0$, i.e., it takes the value 0 when $z \leq 0$ and $\infty$ otherwise; the condition $s = \sigma^2$ is sufficient, but not necessary for constraint satisfaction (Kervadec et al., 2022). This convergence to the characteristic function is visually depicted in Figure 12, showing the change in the log barrier function as we gradually decrease $\sigma$.

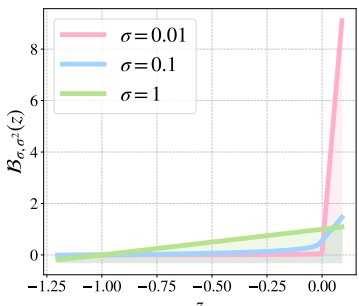

*Figure 12.* **The relaxed logarithmic barrier.** We depict the convergence of the relaxed logarithmic barrier $\mathcal{B}_{\sigma,\sigma^2}(z)$ to the characteristic function $\chi\{z \leq 0\}$ as $\sigma \to 0$. We gradually decrease $\sigma$ from 1 to 0.01. Consequently, $\mathcal{B}_{\sigma,\sigma^2}(z)$ gets closer to 0 for $z \leq 0$ and increases to $\infty$ otherwise.

In Section 3, we had defined

$$\mathcal{L}_{\mathrm{LB}}(\rho, \boldsymbol{\sigma}) = \mathcal{F}(\rho) - \sum_{k=1}^{K-1} \mathcal{B}_{\boldsymbol{\sigma}_k, \boldsymbol{\sigma}_k^2}(\boldsymbol{b}_k - \boldsymbol{\mathcal{G}}_k(\rho)) . \tag{22}$$

Since the log barrier converges to the characteristic function as $\sigma \to 0$, we want to find the maximizer of $\mathcal{L}_{\mathrm{LB}}(\rho, \boldsymbol{\sigma})$ for small $\boldsymbol{\sigma}$. However, doing so directly leads to instabilities as the objective function is ill-conditioned. Instead, it is common practice to follow an iterative procedure: one finds the maximizer for a fixed $\boldsymbol{\sigma}$, reduces $\boldsymbol{\sigma}$, and repeats (Curtis et al., 2024). Specifically, the procedure is instantiated with initial values $\rho_0$, $\boldsymbol{\sigma}_0$, and $0 < \boldsymbol{\gamma}_k < 1$ for $\boldsymbol{\gamma} = \{\boldsymbol{\gamma}_k\}_{k=1}^{K-1}$. On the $t$-th iteration, $\boldsymbol{\sigma}_k^{(t)} \leftarrow \boldsymbol{\gamma}_k \boldsymbol{\sigma}_k^{(t-1)}$ is reduced and $\rho^{(t)} \leftarrow \operatorname{argmax}_\rho \mathcal{L}_{\mathrm{LB}}(\rho, \boldsymbol{\sigma})$ (with initialization $\rho^{(t-1)}$). In doing so, the constraints are gradually enforced, nudging the LLM to satisfy them over the optimization procedure while avoiding instabilities. As $\{\boldsymbol{\sigma}^{(t)}\} \searrow 0$, the weights $\{\rho^{(t)}\}$ converge to the maximizer of the constrained problem.

It is impossible to maximize $\mathcal{L}_{\mathrm{LB}}(\rho, \boldsymbol{\sigma})$ exactly in many practical applications. Instead, at each iteration, one can take a single optimization step toward the solution. Doing so is amenable to stochastic gradient methods and mitigates computational overhead: the optimization proceeds as normal while the value of $\boldsymbol{\sigma}$ is reduced over the course of the procedure. One can

guarantee the convergence of this procedure to the optimal solution in some settings; for example, (Curtis et al., 2024) prove convergence when dealing with box constraints. However, convergence in a scenario like ours is not guaranteed. Nevertheless, we will experimentally demonstrate its use for our constrained problems.

We employ stochastic gradient methods and derive the gradient of our objective function directly:

$$\partial_\rho \mathcal{L}_{\text{LB}}(\rho, \boldsymbol{\sigma}) = \partial_\rho \mathcal{F}(\rho) \; + \; \sum_{k=1}^{K-1} \frac{\boldsymbol{\sigma}_k \partial_\rho \boldsymbol{\mathcal{G}}_k(\rho)}{\max\left(\boldsymbol{\mathcal{G}}_k(\rho) - \boldsymbol{b}_k, \boldsymbol{\sigma}_k^2\right)}. \tag{23}$$

This follows immediately from Equation (22). See (Schulman et al., 2016) for a more detailed review.

**Connection to Lagrange multipliers.** The log barrier and the Lagrangian are intrinsically connected; this becomes evident when comparing Equation (23) with the (gradient of the) Lagrangian in Equation (3). In particular, we define the multipliers:

$$\boldsymbol{\lambda}_k \; = \; \frac{\boldsymbol{\sigma}_k}{\max\left(\boldsymbol{\mathcal{G}}_k(\rho) - \boldsymbol{b}_k, \boldsymbol{\sigma}_k^2\right)} \; .$$

They can be interpreted as Lagrange multipliers: for active constraints, $\boldsymbol{\lambda}_k = 1/\boldsymbol{\sigma}_k$ is non-zero; for inactive constraints, $\boldsymbol{\lambda}_k = \boldsymbol{\sigma}_k / (\boldsymbol{\mathcal{G}}_k(\rho) - \boldsymbol{b}_k)$ vanishes to 0 as $\boldsymbol{\sigma}_k \to 0$. Hence, the KKT complementary slackness condition is satisfied by design.

### A.5. Main Text Proofs

**Proposition A.5.1.** *The dual formulation of Problem (2) is given by,*

$$\begin{aligned}
\mathtt{Dual}^\star &\triangleq \min_{\boldsymbol{\lambda} \geq 0} \max_\rho \; \mathcal{L}(\rho, \boldsymbol{\lambda}) = \min_{\boldsymbol{\lambda} \geq 0} \mathcal{L}(\rho_{\boldsymbol{\lambda}}^\star, \boldsymbol{\lambda}) \\
&= \min_{\boldsymbol{\lambda} \geq 0} \mathcal{F}(\rho_{\boldsymbol{\lambda}}^\star) - \boldsymbol{\lambda}^T (\boldsymbol{b} - \boldsymbol{\mathcal{G}}(\rho_{\boldsymbol{\lambda}}^\star)) \quad , \text{where}, \;\; \forall \, y \in \mathcal{Y}, \\
\rho_{\boldsymbol{\lambda}}^\star(y) &= \exp\left(\tau^{-1} \underset{y' \sim \mu}{\mathbb{E}} [p(y \succ y') + \boldsymbol{\lambda}^T \boldsymbol{q}(y \succ y')] - 1\right) .
\end{aligned} \tag{24}$$

*Proof.* Given the Lagrangian $\mathcal{L}(\rho, \boldsymbol{\lambda})$, the dual formulation is given by

$$\max_\rho \min_{\boldsymbol{\lambda} \geq 0} \mathcal{L}(\rho, \boldsymbol{\lambda}) \equiv \min_{\boldsymbol{\lambda} \geq 0} \max_\rho \mathcal{L}(\rho, \boldsymbol{\lambda}) .$$

By the strong duality, it is sufficient to consider KKT conditions for $(\rho^\star, \boldsymbol{\lambda}^\star)$.

(i) Primal feasibility i.e. $\mathbb{E}_{y \sim \pi_{\text{ref}}, y' \sim \mu}[\rho^\star(y) \, \boldsymbol{q}(y \succ y')] \geq \boldsymbol{b}$.

(ii) Dual feasibility i.e. $\boldsymbol{\lambda}^\star \geq \boldsymbol{0}$.

(iii) Complementary slackness i.e. $(\boldsymbol{\lambda}^\star)^T \left(\boldsymbol{b} - \mathbb{E}_{y \sim \pi_{\text{ref}}, y' \sim \mu}[\rho^\star(y) \, \boldsymbol{q}(y \succ y')]\right) = \boldsymbol{0}$.

(iv) Stationarity i.e. $\nabla_\rho \mathcal{L}(\rho, \boldsymbol{\lambda}) = 0$ i.e.

$$\underset{y' \sim \mu}{\mathbb{E}} [p(y \succ y')] - \tau\left(\ln(\rho(y)) + 1\right) + \boldsymbol{\lambda}^T \underset{y' \sim \mu}{\mathbb{E}} [\boldsymbol{q}(y \succ y')] = 0 .$$

$$\Rightarrow \; \rho^\star(y) = \exp\left(\frac{1}{\tau} \underset{y' \sim \mu}{\mathbb{E}} \left[p(y \succ y') + \boldsymbol{\lambda}^T \boldsymbol{q}(y \succ y')\right] - 1\right)$$

Now we show that conditions (i)-(iii) hold for the above $\rho^\star(y)$. For condition (i), by initialization of $\boldsymbol{b}$ using Proposition A.3.3, we have

$$\underset{\substack{y \sim \pi_{\text{ref}} \\ y' \sim \mu}}{\mathbb{E}} [\rho^\star(y) \, \boldsymbol{q}(y \succ y')] \geq \underset{\substack{y \sim \pi_{\text{ref}} \\ y' \sim \mu}}{\mathbb{E}} [\rho_0(y) \, \boldsymbol{q}(y \succ y')] \geq \boldsymbol{b} \; ,$$

where $\rho_0(y) = \pi_0(y)/\pi_{\text{ref}}(y)$, $\pi_0$ is the initialization point of solving Problem (2), and the second inequality follows by construction. Condition (ii) also holds by construction of $\boldsymbol{\lambda}$. Now, condition (iii) holds by definition if the constraint is active i.e. $\mathbb{E}_{y\sim\pi_{\text{ref}},y'\sim\mu}[\rho^\star(y)\,\boldsymbol{q}(y \succ y')] = \boldsymbol{b}$, and if it is inactive, then dual feasibility also ensures that complimentary slackness holds. As a consequence, all KKT conditions are always satisfied with the above $\rho^\star(y)$, which concludes the proof. $\qquad\square$

**Proposition A.5.2.** *The optimal solution to Problem* (5) *can be obtained by solving the following optimization problem.*

$$\chi_k^\star = \underset{\chi_k \geq 0}{\operatorname{argmax}}\ \mathcal{L}_k(\chi_k\,;\rho)$$

$$:= -\chi_k \ln\left(\mathbb{E}_{y\sim\pi_{\text{ref}},y'\sim\mu}\left[\exp\left(-\chi_k^{-1}\rho(y)\boldsymbol{q}_k(y \succ y')\right)\right]\right) - \chi_k\epsilon$$

$$\text{with,}\ \ \pi_k^\star(y) \propto \pi_{\text{ref}}(y)\underbrace{\exp\left(-(\chi_k^\star)^{-1}\underset{y'\sim\mu}{\mathbb{E}}\left[\rho(y)\boldsymbol{q}_k(y \succ y')\right]\right)}_{w(y)\ \textit{(unnormalized weight)}}.$$

*Proof.* For the given constrained optimization problem:

$$\min_{\pi_k}\ \underset{\substack{y\sim\pi_k \\ y'\sim\mu}}{\mathbb{E}}\left[\rho(y)\boldsymbol{q}_k(y \succ y')\right]\ \ \texttt{s.t.}\ \ \mathsf{KL}(\pi_k \,\|\, \pi_{\text{ref}}) \leq \epsilon\ \ \text{and,}\ \ \sum_{y\in\mathcal{Y}}\pi_k(y) = 1\,,$$

we consider its Lagrangian to find its dual problem. By noticing that, $\pi_k(y)\rho(y) = \pi_k(y)\frac{\pi(y)}{\pi_{\text{ref}}(y)} \approx \pi(y)$, the Lagrangian for some arbitrary multipliers $\chi_k \geq 0$ and $\zeta_k \in \mathbb{R}$ is given by,

$$\mathcal{L}(\pi_k,\chi_k,\zeta_k) = \underset{\substack{y\sim\pi_k \\ y'\sim\mu}}{\mathbb{E}}\left[\rho(y)\boldsymbol{q}_k(y \succ y')\right] + \chi_k\left(\underset{y\sim\pi_k}{\mathbb{E}}\left[\ln\left(\frac{\pi_k(y)}{\pi_{\text{ref}}(y)}\right)\right] - \epsilon\right) + \zeta_k\left(\sum_{y\in\mathcal{Y}}\pi_k(y) - 1\right)$$

$$= \underset{\substack{y\sim\pi_k \\ y'\sim\mu}}{\mathbb{E}}\left[\rho(y)\boldsymbol{q}_k(y \succ y')\right] + \chi_k\left(\sum_{y\in\mathcal{Y}}\pi_k(y)\ln\left(\frac{\pi_k(y)}{\pi_{\text{ref}}(y)}\right) - \epsilon\right) + \zeta_k\left(\sum_{y\in\mathcal{Y}}\pi_k(y) - 1\right),$$

where $\chi_k \in \mathbb{R}_+$ is the Lagrange multiplier for KL constraint, and $\zeta_k \in \mathbb{R}$ is the Lagrange multiplier for the normalization constraint that ensures $\sum_{y\in\mathcal{Y}}\pi_k(y) = 1$. Hence, the corresponding optimization problem due to strong duality (Slater's conditions) is: $\min_{\pi_k}\max_{\chi_k\geq 0,\zeta_k}\mathcal{L}(\pi_k,\chi_k,\zeta_k) \equiv \max_{\chi_k\geq 0,\zeta_k}\min_{\pi_k}\mathcal{L}(\pi_k,\chi_k,\zeta_k)$. Now, we can compute the non-parametric closed form solution for each sample $y$ for the inner minimization problem. Due to the convexity of KL-divergence, it is sufficient to consider $\nabla_{\pi_k}\mathcal{L}(\pi_k,\chi_k,\zeta_k) = 0$. For each $y$ we then have,

$$\nabla_{\pi_k}\mathcal{L}(\pi_k,\chi_k,\zeta_k) = \underset{y'\sim\mu}{\mathbb{E}}\left[\rho(y)\boldsymbol{q}_k(y \succ y')\right] + \chi_k^\star\left(\ln\left(\frac{\pi_k^\star(y)}{\pi_{\text{ref}}(y)}\right) + 1\right) + \zeta_k = 0$$

$$\Rightarrow\ \pi_k^\star(y) \propto \pi_{\text{ref}}(y)\exp\left(-(\chi_k^\star)^{-1}\underset{y'\sim\mu}{\mathbb{E}}\left[\rho(y)\boldsymbol{q}_k(y \succ y')\right]\right)$$

with some normalization constant $Z_k$ that ensures that $\sum_{y\in\mathcal{Y}}\pi_k^\star(y) = 1$, which is described with respect to $\zeta_k$. Then, by plugging the above stationarity condition into the Lagrangian, we have the dual unconstrained optimization problem as,

$$\max_{\chi_k\geq 0,\zeta_k}\mathcal{L}(\pi_k^\star,\chi_k,\zeta_k) = \underset{\substack{y\sim\pi_k^\star \\ y'\sim\mu}}{\mathbb{E}}\left[\rho(y)\boldsymbol{q}_k(y \succ y')\right] + \chi_k\left(\underset{y\sim\pi_k^\star}{\mathbb{E}}\left[\ln\left(\frac{\pi_k^\star(y)}{\pi_{\text{ref}}(y)}\right)\right] - \epsilon\right) + \zeta_k\left(\sum_{y\in\mathcal{Y}}\pi_k^\star(y) - 1\right)$$

$$= \underset{y\sim\pi_k^\star}{\mathbb{E}}\left[\underset{y'\sim\mu}{\mathbb{E}}\left[\rho(y)\boldsymbol{q}_k(y \succ y')\right] + \chi_k\ln\left(\frac{\pi_k^\star(y)}{\pi_{\text{ref}}(y)}\right)\right] - \chi_k\epsilon$$

$$= -\chi_k\ln\left(\mathbb{E}_{y\sim\pi_{\text{ref}},y'\sim\mu}\left[\exp\left(-\chi_k^{-1}\rho(y)\boldsymbol{q}_k(y \succ y')\right)\right]\right) - \chi_k\epsilon.$$

This concludes the proof. $\qquad\square$

### A.6. Empirical Optimization Problem

We now formulate the problem given a fixed offline dataset $\mathcal{D}$ of the form $\mathcal{D} := \{(x_i, y_i, y_i', \mathbb{I}(y_i, y_i'))\}_{i=1}^N$, where context $x_i \in \mathcal{X}$, $y_i, y_i'$ are two generations from $\pi_{\text{ref}}$ and $\mu$ respectively, and $\mathbb{I}(\cdot, \cdot) \in \{0,1\}^K$ is preference indicator vector over $K$ objectives, i.e., $\mathbb{I}_k(y, y') = 1$ if $y \succ_k y'$, and $0$ otherwise for $k \in [K]$. With a slight abuse in notation, we let $\mathbb{I}_p(\cdot, \cdot) \triangleq \mathbb{I}_K(y, y')$ and $\mathbb{I}_q(\cdot, \cdot) \triangleq (\mathbb{I}_1(y, y'), \dots, \mathbb{I}_{K-1}(y, y'))^T$. Then, the empirical optimization problem is given by:

$$\max_\rho \underbrace{\frac{1}{N} \sum_{i=1}^N \rho(y_i) \, \mathbb{I}_p(y_i, y_i') - \tau \rho(y_i) \ln(\rho(y_i))}_{\widehat{\mathcal{F}}(\rho)} \quad \text{s.t.} \quad \underbrace{\frac{1}{N} \sum_{i=1}^N \rho(y_i) \, \mathbb{I}_q(y_i, y_i') \geq \boldsymbol{b}.}_{\widehat{\boldsymbol{\mathcal{G}}}(\rho)} \tag{25}$$

*Proof.* We need to show that we can build an unbiased estimate of the optimization function from empirical observations. To this end, consider the sampled COP as:

$$\max_\rho \mathop{\mathbb{E}}_{\substack{y \sim \pi_{\text{ref}} \\ y' \sim \mu}} [\rho(y) \, I_p(y, y')] - \tau \mathop{\mathbb{E}}_{\substack{y \sim \pi_{\text{ref}} \\ y' \sim \mu}} [\rho(y) \ln(\rho(y))] \quad \text{s.t.} \quad \mathop{\mathbb{E}}_{\substack{y \sim \pi_{\text{ref}} \\ y' \sim \mu}} [\rho(y) \, \boldsymbol{I}_q(y, y')] \geq \boldsymbol{b} \tag{26}$$

, where $\boldsymbol{I}(\cdot, \cdot)$ is a Bernoulli random preference vector over $K$ objectives i.e. $\boldsymbol{I}_k(y, y')$ is a random variable sampled from a Bernoulli distribution with mean $p_k(y \succ y')$ such that it is $1$ if $y \succ_k y'$, and $0$ otherwise, where $\succ_k$ denotes preference with respect to the $k^{th}$ objective for $k \in [K]$. Following the notation discussed before, we let $I_p(\cdot, \cdot)$ to be the preference with respect to the $K^{th}$ objective, and let $\boldsymbol{I}_q(\cdot, \cdot)$ to be the preference vector for the remaining $(K-1)$ objectives i.e. $I_p(y, y') = \boldsymbol{I}_K(y, y')$ and $(\boldsymbol{I}_q)_k = \boldsymbol{I}_k(y, y')$ for $k \in [K-1]$. Note that Problem (2) and Problem (26) are equivalent since $\mathbb{E}_{y \sim \pi_{\text{ref}}, y' \sim \mu}[\rho(y) \, I_p(y, y')] = \mathbb{E}_{y \sim \pi_{\text{ref}}, y' \sim \mu}[\rho(y) \, I_p(y, y') \,|\, y, y'] = \mathbb{E}_{y \sim \pi_{\text{ref}}, y' \sim \mu}[\rho(y) \, p(y \succ y')]$. Similar argument follows for $\boldsymbol{I}_q$. $\qquad\square$

Note that each data point $(x_i, y_i, y_i', \mathbb{I}(y_i, y_i'))$ contributes two terms to the empirical problem above: one with $(x, y, y', \mathbb{I}(y, y')) = (x_i, y_i, y_i', \mathbb{I}(y_i, y_i'))$ and another with $(x, y, y', \mathbb{I}(y, y')) = (x_i, y_i', y_i, \mathbb{I}(y_i', y_i))$. This symmetry is important to exploit since it reduces gradient variance and improves stability during optimization. For clarity, we omit the symmetric term in notation as incorporating it is trivial – simply augment the current dataset by swapping $y_i$ and $y_i'$ and bit flipping $\mathbb{I}(\cdot, \cdot)$ element-wise.

Now, as before, for some $\boldsymbol{\lambda} := \{\lambda_k\}_{k=1}^{K-1} \geq \boldsymbol{0}$, we have the Lagrangian as $\widehat{\mathcal{L}}(\rho, \boldsymbol{\lambda}) = \widehat{\mathcal{F}}(\rho) - \boldsymbol{\lambda}^T \left(\boldsymbol{b} - \widehat{\boldsymbol{\mathcal{G}}}(\rho)\right)$, and the dual as,

$$\widehat{\texttt{Dual}^\star} \triangleq \min_{\boldsymbol{\lambda} \geq \boldsymbol{0}} \max_\rho \widehat{\mathcal{L}}(\rho, \boldsymbol{\lambda}) = \min_{\boldsymbol{\lambda} \geq \boldsymbol{0}} \widehat{\mathcal{L}}(\rho_{\boldsymbol{\lambda}}^\star, \boldsymbol{\lambda}) = \min_{\boldsymbol{\lambda} \geq \boldsymbol{0}} \widehat{\mathcal{F}}(\rho_{\boldsymbol{\lambda}}^\star) - \boldsymbol{\lambda}^T (\boldsymbol{b} - \widehat{\boldsymbol{\mathcal{G}}}(\rho_{\boldsymbol{\lambda}}^\star))$$

$$\text{where,} \quad \rho_{\boldsymbol{\lambda}}^\star(y) = \exp\left((\tau N)^{-1} \sum_{i=1}^N [\mathbb{I}_p(y, y') + \boldsymbol{\lambda}^T \mathbb{I}_q(y, y')] - 1\right). \tag{27}$$

Following our earlier discussion on lower bounding the preference probabilities for $[K-1]$ constraints, we wish to constrain $\texttt{LowerBound}(\widehat{\boldsymbol{\mathcal{G}}}(\rho))$ to be greater than $\boldsymbol{b}$. Then, following Equation (8), the empirical lower bound is obtained by solving below for $M$ batches of $\mathcal{D}$, where each batch $m$ of size $N_M = \lfloor N/M \rfloor$ is of the form $\left(x_{m,j}, y_{m,j}, y_{m,j}', \mathbb{I}(y_{m,j}, y_{m,j}')\right)_{j=1}^{N_M}$.

$$\min_{\boldsymbol{\chi} \geq \boldsymbol{0}} \widehat{J}(\boldsymbol{\chi}; \rho) := \frac{1}{M} \sum_{m=1}^M \left[ \sum_{k=1}^{K-1} \left[ \chi_k \ln\left(\frac{1}{N_M} \sum_{j=1}^{N_M} \exp\left(\chi_k^{-1}[\rho(y_{m,j})(\mathbb{I}_q)_k(y_{m,j}, y_{m,j}')]\right)\right) + \chi_k \epsilon \right]\right]. \tag{28}$$

This transforms the empirical dual Problem (27) into the below optimizations for $\boldsymbol{\lambda}$ and policy $\pi_\psi$:

$$\min_{\boldsymbol{\lambda} \geq \boldsymbol{0}} \widehat{J}(\boldsymbol{\lambda}; \boldsymbol{\chi}) := \widehat{\mathcal{F}}(\rho) - \boldsymbol{\lambda}^T (\boldsymbol{b} - \widehat{J}(\boldsymbol{\chi}; \rho)) \quad \text{and,} \quad \min_\psi \widehat{J}_\rho(\pi_\psi) := -\frac{1}{N} \sum_{i=1}^N \rho_{\boldsymbol{\lambda}}^\star(y_i) \log(\pi_\psi(y_i)), \tag{29}$$

where $\rho_{\boldsymbol{\lambda}}^\star(\cdot)$ is computed using Equation (27).

## A.7. Implementation Details

### A.7.1. BACKGROUND

**SFT.** Supervised fine-tuning (SFT) with labeled demonstrations is widely adopted to fine-tune LLMs (Zhang et al., 2023; Peng et al., 2023). Given prompt-response pairs $\{(x, y)\}$ sampled form the dataset $\mathcal{D}$, the SFT loss function is defined as:

$$\mathcal{L}_{\text{SFT}} = -\mathbb{E}_{(x,y)\sim\mathcal{D}}\left[\sum_i \log \pi_{\text{sft}}(y_i|x, y_{<i})\right], \tag{30}$$

where $\pi_{\text{sft}}$ refers to the LLM policy and $y_{<i}$ indicates all tokens before the $i$-th token in response $y$.

**RLHF.** RLHF typically involves two steps (Ouyang et al., 2022; Wu et al., 2023): reward modeling, and RL training. In reward modeling, a reward model $r_\phi$ is trained to minimize the loss function $\mathcal{L}_{\text{RM}}(\phi) = -\mathbb{E}_{(x,y_w,y_l)\sim\mathcal{D}}[\log(\sigma(r_\phi(x, y_w) - r_\phi(x, y_l)))]$, where $\sigma(z)$ is the sigmoid function, $y_w$ and $y_l$ refer to preferred and dispreferred responses, respectively. Generally, RL training uses the PPO algorithm (Schulman et al., 2017) with an additional KL penalty relative to the SFT policy:

$$\arg\max_{\pi_\theta}\mathbb{E}_{x\sim\mathcal{D},y\sim\pi_\theta(y|x)}\left[r_\phi(x, y) - \tau \log \frac{\pi_\theta(y|x)}{\pi_{\text{sft}}(y|x)}\right],$$

where $\tau > 0$ is the KL penalty coefficient.

### A.7.2. TRAINING DETAILS.

We summarize the key implementation details of text generation tasks in Table 3. This table also provides links to the open-sourced datasets and reward models utilized in our study. Implementation is primarily based on trl (von Werra et al., 2020b). Especially, SFT fine-tunes the base model, while MORLHF and Rewarded Soups fine-tune the SFT model using the PPO algorithm. In contrast, RiC directly fine-tunes the base model. See (Yang et al., 2024) for more details. We apply the same 4-bit quantization and LoRA configuration for training all models. During evaluation, we maintain a consistent configuration across different models, generating 64 tokens for the Helpful Assistant task and 32 for the Reddit Summary task.

For all baselines including MOPO, we begin by normalizing the rewards using the mean and standard of the offline dataset before incorporating them into the prompts. During online generation (updating the reference policy) and evaluation, we sample a group of 25,000 random samples from a normal distribution and use the maximum and minimum values (generally around $\pm 3$) of these samples to replace the maximum and minimum values of the dataset. This method can prevent the extreme values in the dataset to impact reward selection.

**Incorporating preference vectors.** Pareto fronts are generated as in (Yang et al., 2024). One point to consider while evaluating empirical Pareto fronts is to incorporate user preferences for a particular objective. For instance, in the case of 2 objectives, in RiC (Yang et al., 2024), scalarization tuples are passed as in-context human preferences. Preference tuple $w = (w_1, w_2)$ for the two reward dimensions is passed to the model at inference time to adjust the LLM policy according to the user preferences. It is necessary to map these scalarization tuples $w$ to the desired rewards that will be used as conditioning in prompts. Similarly, for the two objective case, PARM uses prompts at inference time and clusters them. Then, they take two diverse policies, apply each policy to each cluster, and compute a multi-objective reward vector for all policy-cluster pairs. A single point in the plots is a reward vector for the policy-cluster pair. For a fair comparison, MOPO should also incorporate the preferences at inference time (in terms of which objective should be primary). For $K$ objectives, one idea would be to solve $K$ separate optimization problems as in Problem (2), one for each objective being the "main" objective. However, this is not efficient since if the LLM had $m$ parameters, the space complexity of representing $K$ policies would be $Km$.

We propose a multi-policy architecture in which all $K$ policies share a common LLM backbone. Each policy $k$ is parameterized by a matrix $\theta_k \in \mathbb{R}^{L\times d}$, where $L$ is the number of tokens and $d$ is the transformer embedding dimension. The logit for the next token under policy $k$ is computed as $\theta_k\phi$, where $\phi \in \mathbb{R}^d$ is the final-layer embedding summarizing the input sequence. This multi-headed model is illustrated in Figure 13. The total parameter count is reduced to $m - dL + dKL$, where $m - dL$ parameters are in the shared backbone, and $dKL$ correspond to the $K$ policy heads. The shared backbone enables efficient language modeling, while the separate heads provide sufficient flexibility for each policy to optimize distinct objectives and language styles. During training, each policy matrix is jointly trained with the transformer backbone.

At inference time, the user specifies which objective $k_0 \in [K]$ should be the primary objective, and the model computes logit for the text token using $\theta_{k_0} \phi$ to adapt to inference time user specifications.

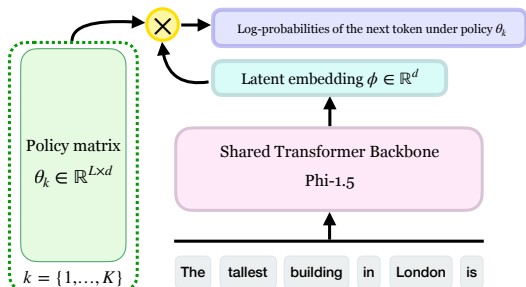

*Figure 13.* Multi-headed policy architecture for incorporating preferences at inference time in `MOPO`.

**Inference code.** Here we provide the inference pseudo-code. Notably, to prevent potential precision explosion, we approximate the solution for JSD same as Reverse KL-divergence, as they are inherently similar.

```python
def f_divergence(logp, weights, f_type):
    if f_type in ("reverse_kld", "jsd"):
        return torch.stack([w * lp for w, lp in zip(weights, logp)]).sum(dim=0)

    if f_type == "forward_kld":
        alpha = 1.0
    elif "-divergence" in f_type:
        alpha = float(f_type.split("-", 1)[0])
    else:
        raise ValueError(f"Unknown_f_type:_{f_type}")

    terms = [
        -alpha * lp + np.log(w)
        for w, lp in zip(weights, logp)
        if w != 0
    ]
    return -torch.logsumexp(torch.stack(terms), dim=0)
```

**Compute resources.** Average training times for the 7b parameter family of LLMs (OpenChat-v3.5, Llama-3.1-8B, Mistral-7b-v0.2 (Instruct), and Zephyr-7b-beta) are provided. For training RLHF and MODPO models, the number of workers are set as 3, each taking up $7,000$M of memory, running for $5.1$ and $5.7$ hours respectively; for training RiC and PARM models, the number of workers are set as 2, each taking up $11,000$M of memory, running for $3.4$ and $3.2$ hours respectively. For `MOPO`, see Table 3, which takes $4.1$ hours.

**Training hyper-parameters.** For PPO, we follow the settings of (Yang et al., 2024) and train for 25 batches; for DPO, we follow (Zhou et al., 2023), with PERDEVICE_BATCH_SIZE$= 1$ and MAX_LENGTH$= 64$.

**Inference hyper-parameters.** For PPO, we follow the settings of (Yang et al., 2024) with NUM_BEAMS$= 1$; for DPO, we follow (Zhou et al., 2023) with BATCH_SIZE$= 4$, MAX_LENGTH$= 50$ and NUM_BEAMS$= 1$.

**Codebase.** Our codebase is mainly based on trl (von Werra et al., 2020a) (https://github.com/huggingface/trl), MODPO (https://github.com/ZHZisZZ/modpo), RiC (https://github.com/YangRui2015/RiC) and Finegrained RLHF (https://github.com/allenai/FineGrainedRLHF), and has referred to f-divergence DPO (Wang et al., 2024a) (https://github.com/alecwangcq/f-divergence-dpo), Pack-LLM (Mavromatis et al., 2024) (https://github.com/cmavro/PackLLM), and DPA (Wang et al., 2024b) (https://github.com/Haoxiang-Wang/directional-preference-alignment).

### A.8. Ablation Study (cont.)

In Section 4 we briefly discussed the impact of hyperparameters on the `MOPO` optimization procedure. Here we present complete results of hyperparameters. In addition to constrain relaxation and lagged policy update parameters, we show results of regularization as well in Figure 14.

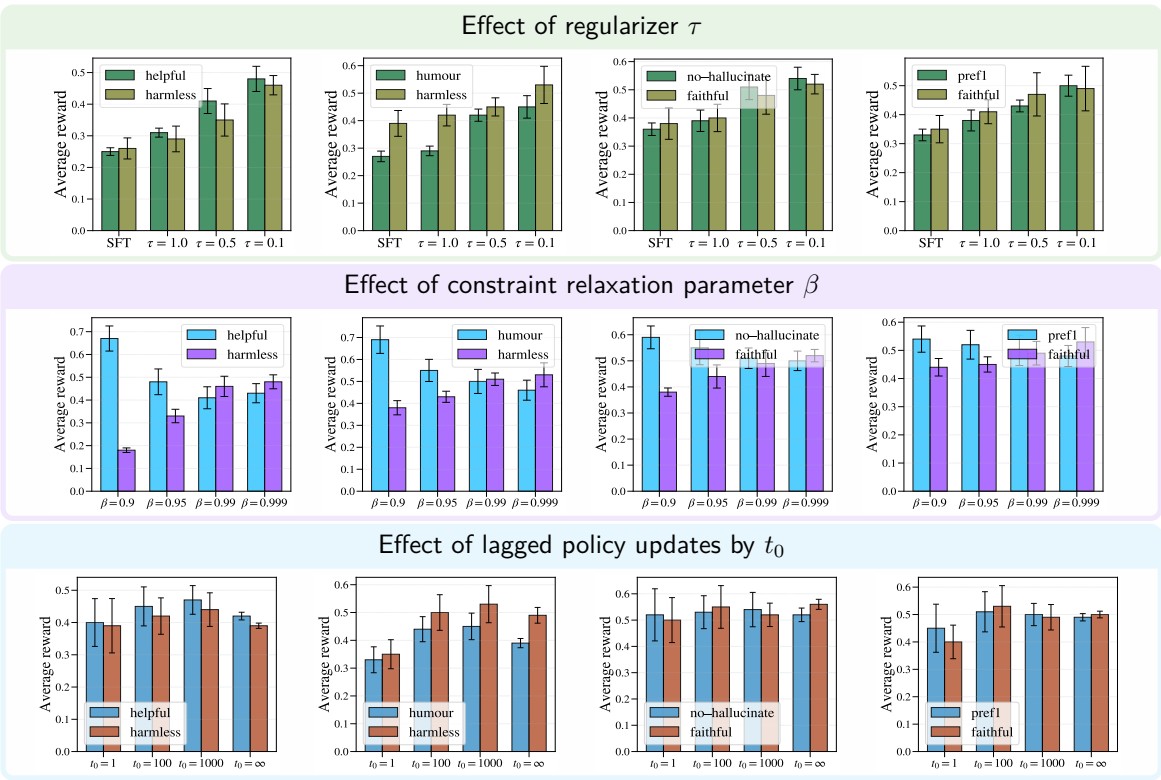

*Figure 14.* Normalized rewards of Llama-3.1-8B aligned using `MOPO` under varying optimization parameters on the Helpful Assistant and Reddit Summary tasks.

### A.9. Example Outputs

Example generations for each dataset are shown in Tables 4 and 5. For each dataset, we show a representative prompt in the down-sampled dataset, and one generated response for each model/algorithm, with preference weightings set as $w = (0.5, 0.5)$ for MODPO, PARM, and RiC.

### A.10. Three-objective Experimental Results

Here we present results for the three-objective text generation tasks in Table 6.

*Table 3.* Key implementations of the text generation experiments.

| Basic information | |
|---|---|
| Architecture | Transformer |
| Pre-training (SFT) | See Section 4 |
| Hardware | NVIDIA A100 80GB, 1 accelerator, 12 vCPU |
| Quantization for training | 4bit |
| Fine-tuning strategy | LoRA (Hu et al., 2022a) |
| LoRA $r$ | 16 |
| LoRA alpha | 32 |
| LoRA dropout | 0.05 |
| Optimizer | Adam |
| Batch size | 8 |
| Inference tokens for evaluation | 64 for Helpful Assistant and 32 for Reddit Summary |

| SFT | |
|---|---|
| Finetuning steps | 10000 |
| Initial learning rate | 1.41e-4 |
| Learning rate scheduler | Linear |

| MOPO | |
|---|---|
| Finetuning steps | 10000 |
| Initial learning rate | 1.87e-4 |
| Learning rate scheduler | Linear |
| Batch size | 8 |
| Regularization $\tau$ | 0.08 |
| Constraint lower bound ball $\epsilon$ | 0.15 |
| Constraint relaxation $\beta$ | 0.9995 |
| Reference policy lag $t_0$ | 500 |

| RiC | |
|---|---|
| Offline finetuning steps | 10000 |
| Initial learning rate | 1.41e-4 for offline finetuning, 1e-5 for online finetuning |
| Learning rate scheduler | Linear for offline finetuning, constant for online finetuning |
| Threshold for MORS | 0.7-quantile for each reward dimension |
| Online generation sample size per iteration | 10000 |
| Online finetuning steps per iteration | 4000 |

| RL step for MODPO | |
|---|---|
| RL algorithm | PPO (Schulman et al., 2017) |
| Implementation | trl (von Werra et al., 2020b) |
| KL regulaization | 0.2 |
| Epochs | 1 |
| learning rate | 1e-5 |
| lambda for GAE | 0.95 |
| gamma | 1 |
| cliprange | 0.2 |
| Number of optimisation epochs per batch | 4 |
| Target KL | 3 |

| Datasets and Reward Models | |
|---|---|
| Task name | **Helpful Assistant** |
| Description | Provide helpful and harmless answers to potentially complex and sensitive questions. |
| Prompt | No prompt, only users' questions. |
| Dataset | Anthropic/hh-rlhf (Bai et al., 2022) |
| harmless reward | gpt2-large-harmless-reward_model |
| helpful reward | gpt2-large-helpful-reward_model |
| humor reward | humor-no-humor |
| Task name | **Reddit Summary** |
| Description | Provide a summary to a post from Reddit. |
| Prompt | Generate a one-sentence summary of this post. |
| Dataset | openai/summarize_from_feedback (Stiennon et al., 2020) |
| pref1 reward | gpt2_reward_summarization |
| less-hallucination reward | bart-summary-detector |
| faithful reward | bart-faithful-summary-detector |

*Table 4.* Examples of **Reddit Summary**.

| Prompt: | **Instruction:** Rewrite the email below in a friendly, concise tone in a single sentence.
**Input:** Dear team, due to updated budget constraints, we must reduce remote work days from three to one per week effective next month. Please ensure on-site attendance Monday–Thursday. Thank you for your cooperation.
**Response:** |
|---|---|
| **LM response** | |
| **MODPO:** | Starting next month, please work onsite Monday–Thursday; budget changes limit us to one remote day weekly. |
| **PARM:** | Budget limits mean we'll shift to one WFH day—please join us on-site Mon–Thu from next month; thanks for flexing with us! |
| **RiC:** | Heads-up: from next month we'll be in the office Mon–Thu (just one WFH day) because of budget constraints—appreciate everyone's flexibility! |
| MOPO: | Hey team! Budget tweaks mean we'll have one remote day going forward, so plan to be on-site Mon–Thu starting next month—thanks for rolling with the change! |

*Table 5.* Examples of **Helpful Assistants**. Helpfulness & Humor.

| Prompt:
Assistant: | **Human:** What is two factor authentication, and how can it help me stay safe online? |
|---|---|
| **LM response** | |
| **RLHF (Helpful):** | Two factor authentication is a security system that requires you to provide two pieces of information to verify your identity. This can help protect you from fraud and identity theft. |
| **RLHF (Humor):** | What do you mean by two factor authentication? |
| **MODPO:** | Two-factor authentication (2FA) adds a second proof—usually a code from your phone—on top of your password, so a thief who steals the password still can't get in. |
| **PARM:** | Think of 2FA as a double lock: you enter your password, then confirm with a phone code or authenticator app. That quick extra step blocks most account-takeovers—using it on email first is a great start! |
| **RiC:** | It's a "prove-it-twice" check—password *plus* something you have (SMS code, hardware key). Turn it on for banking, email, socials and you slam the door on 99% of drive-by hacks. |
| MOPO: | Picture ordering pizza: the driver needs your address *and* a secret knock. 2FA works the same—password first, then a six-digit code or hardware tap. Even if crooks guess the password, the knock stays secret, so enable it everywhere (especially email and banking) for huge peace-of-mind gains. |

*Table 6.* Three-objective alignment for Helpful Assistant task with normalized rewards across different SFT models.

| | phi-1.5 | | | OpenChat-v3.5 | | | Llama-3.1-8B | | | Mistral-7b-v0.2 (Instruct) | | |
|---|---|---|---|---|---|---|---|---|---|---|---|---|
| | helpful | humour | harmless | helpful | humour | harmless | helpful | humour | harmless | helpful | humour | harmless |
| RLHF-r1 | 0.65 | -0.73 | -0.47 | 0.72 | -0.38 | -0.20 | 0.69 | -0.33 | -0.18 | 0.74 | -0.41 | -0.22 |
| RLHF-r2 | -0.93 | 0.44 | -0.53 | -0.78 | 0.49 | -0.37 | -0.80 | 0.50 | -0.39 | -0.79 | 0.52 | -0.38 |
| RLHF-r3 | -0.88 | -0.97 | 0.29 | -0.77 | -0.89 | 0.40 | -0.80 | -0.90 | 0.43 | -0.78 | -0.91 | 0.41 |
| RiC | 0.18 | 0.10 | 0.05 | 0.27 | 0.16 | 0.12 | 0.26 | 0.14 | 0.11 | 0.25 | 0.15 | 0.10 |
| PARM | 0.23 | 0.12 | 0.11 | 0.32 | 0.18 | 0.24 | 0.30 | 0.17 | 0.22 | 0.45 | 0.16 | 0.23 |
| MODPO | -0.04 | -0.17 | 0.00 | 0.05 | -0.08 | 0.09 | 0.03 | -0.10 | 0.07 | 0.04 | -0.09 | 0.08 |
| DPO on $\mathcal{D}_J$ | 0.13 | 0.05 | 0.08 | 0.17 | 0.08 | 0.10 | 0.19 | 0.10 | 0.12 | 0.18 | 0.09 | 0.11 |
| MOPO-LB | 0.24 | 0.13 | 0.14 | 0.29 | 0.18 | 0.17 | 0.31 | 0.20 | 0.19 | 0.30 | 0.19 | 0.18 |
| MOPO-Lag | 0.20 | 0.10 | 0.09 | 0.40 | 0.23 | 0.18 | 0.38 | 0.21 | 0.16 | 0.39 | 0.22 | 0.17 |

