# OpenReview forum: "Multi-Objective Preference Optimization: Improving Human Alignment of Generative Models"
_ICML.cc/2026/Conference — ICML 2026 regular_

### Official Review · Reviewer_fJ8e · 2026-03-13

**Soundness:** 3
**Presentation:** 2
**Significance:** 3
**Originality:** 3
**Overall Recommendation:** 3
**Confidence:** 4

**Summary:**

The paper proposes Multi-Objective Preference Optimization (MOPO) that finds a policy maximizes that a primary objective is maximized (with a entropic regularizer), under the constraint that other objectives lower bounds stay above a threshold. The lower bounds are defined under a KL constraint to keep the policy close to a reference policy. Authors propose a relaxation of this formulation and a Lagrangian objective to optimize in an alternating optimization fashion between finding optimal Lagrangian and optimizing the policy. Experiments in LLM alignment of models such as Phi 1 and models in the 7 B family show the effectiveness of the method.

**Compliance With Llm Reviewing Policy:**

Affirmed.

**Final Justification:**

I thank the authors for providing the additional experiments for fixed temperature with the caveat that the choice of the temperature used was not stated , nor tuned to see how this behaves in practice.  I think the simple baseline should be exercised more and presented in the paper.

**Key Questions For Authors:**

* As stated above the main innovation in the paper is in moving from expected preference for j\neq i , to softmin, defined by the LSE (log sum exp operator). The temperatures of the softmin are learned lagrangians for each objectives, that correspond to balances objectives and constraint on KL. I encourage the author to give those intuitions in the paper that they are effectively moving to softmin, with learned temperature.  A simple baseline is to fix the temperature for all objectives, this will be make the algorithm simpler and easier to digest, can you in your synthetic example show that learned temperature improve upon a simple baseline with fixed temperature? Also how this simple baseline performs in practice? Indeed  \min_{p} E_{p}f(x) + \beta KL (\pi || \pi_{\ref}) = -\beta \log E_{p} e^{-f/\beta} = softmin_{p,\beta} f    , so instead of epsilon "KL ball " you are fixing beta across all objectives $j \neq i$. and this will give a close form immediately instead of an optimization problem to solve for each j \neq i

* One question on the algorithm  since you  constrain your problem with KL, instead of penalizing it. one can see the formulation as a multilevel optimization. I wonder if you thought about what would stabilize the training  : usually you either backpropagate via inner problem, think of using implict function theorem and use hessian of inner problems, do as in GANs , multiple iteration  updates for inner problems. Have you looked if beyond the simple alternating optimization , one can obtain a better algorithm.

* How do you choose b and \epsilon in practice?

* When using softmin , the non linearity in log makes the estimation biased , have you encountered issues needing larger batches so that learning the lagrangian gets stable ? That is why i think it is better to fix beta , this will give you less optimizations to do.

**Limitations:**

- The theoretical analysis of the method is still limited
- More ablation on the optimization as suggested above is needed

**Strengths And Weaknesses:**

**Soundness:** The paper is sound in terms of the derivation of the method , finding the lagrangian and proposing an alternating optimization to  the formulation.

**presentation** The paper would benefit from simplification of the notation , for example  $\mu$ and $\nu$ are not clearly defined, and by looking at equation 2 one would not immediately follow what is being optimized since the notation are fuzzy why don't you just fix a $i$ and in the constraint say it needs to be defined for all j \neq  i. It is the same but it will lower the complexity of parsing b is a vector etc on the reader. The problem setting should be simplified and with better definitions.

**Significance** The paper tackles in a nice way the problem of multiple preference optimization , constraining the lower bound of other objectives is a nice contributions. It remains to know how to choose the treshold b_j , j \neq i, and epsilon that are the hyperparameters of the method.

**Originality** I think the main originality is in moving from expected reward for j \neq i in Eq 2, to softmin  as defined in proposition 3.

---

> ### Author Rebuttal · Authors · 2026-03-31
>
> Thank you very much for a thorough review of our paper. We try to address your concerns below:
>
> - **Strengths And Weaknesses:**
> 1. *Presentation*: Please see Line 128 and Line 148 where $\mu$ and $\nu$ are defined. Regarding notation, thank you for this insightful comment. We will look into incorporating this in the revised version of the paper
> 2. *Significance*: Please see our responses to Reviewer AfB9 (Question 2) and Reviewer SmiK (Question 1). We also provide an adaptive constraint schedule starting from Line 259, which discusses a practical solution to the constraint specification problem. Empirical results in Section 4 show that this approach works well in practice as well.
>
> - **Questions:**
> 1. Thank you for this comment. From what we understand, you are proposing to add a fixed penalty (‘constant temperature Lagrangians’) to the primary objective instead of doing iterative primal-dual updates. If that is the case, we know from core RL policy optimization methods (ACPO [1], CPO [2], RCPO [3]) that simply adding a penalty term is empirically much inferior to primal-dual methods. Nevertheless, we agree with you that adding a discussion and some intuition to the paper will help readers, and we will do so in the revised version. Thank you for the suggestion
> 2. This is a very interesting question. Although the problem is formulated as a minimax optimization over the Lagrangian ($\min\_{\lambda \ge 0} \max\_{\rho} \mathcal{L}(\rho, \lambda)$), the "inner" primal problem for the importance sampling ratio $\rho$ admits a closed form solution $\rho\_{\lambda^{\star}}$ (Proposition 3.1). Because this relationship is explicit and differentiable, the dual objective $J(\lambda) = \mathcal{L}(\rho\_\lambda^*, \lambda)$ can be optimized directly via gradient descent-ascent. This bypasses the need for Implicit Function Theorem or Hessian-based backpropagation, as we are not differentiating through a numerical optimization loop but through an analytical expression. Secondly, training stability is precisely why we introduced the LowerBound optimization (Equation 5). Instead of standard alternating optimization which can be sensitive to noise in offline preference estimates, MOPO explicitly constrains the pessimistic lower bound of the secondary objectives. This acts as a robust regularizer that stabilizes the dual variables $\lambda$, providing more reliable updates than GAN-style heuristics or multi-step inner iterations.
> 3. Please see above
> 4. While we agree that constant temperatures will result in a simpler algorithm, vanilla penalty-based methods are known to be inferior to interior point or primal-dual methods. Please see the section starting on Line 256, where we discuss how we use mini-batch approximations to overcome bias during optimization.
>
> - **Limitations:**
> 1. Please see our response to Reviewer SmiK, where we provide some results briefly. We will include a more detailed version in the revised version.
>
> &nbsp;
>
> Please let us know if you have any more questions, we would be happy to discuss. If you are satisfied with our response, we request you to please reconsider your score. Thank you again for your time and feedback.
>
> &nbsp;
>
> [1] A. Agnihotri, et al. ACPO: A Policy Optimization Algorithm for Average MDPs with Constraints. ICML 2024.
>
> [2] J.  Achiam, et al. Constrained policy optimization. ICML 2017.
>
> [3] C. Tessler, et al. Reward Constrained Policy Optimization. ICLR 2019.

---

> > ### Author Rebuttal · Reviewer_fJ8e · 2026-04-01
> >
> > Thank you for your clarifications, I still think the fixed temperature baseline should be performed in the paper.

---

> > > ### Author Response · Authors · 2026-04-02
> > >
> > > We wish to thank you for engaging in discussion with us. We conducted additional experiments to help clear your concerns. As per your suggestion, we compare MOPO-Lag and MOPO-LB  with a **f**ixed **t**emperature baseline called MOPO-FT. We ran experiments on Llama-3.1-8B and Qwen3-8B, and present the results below.
> > >
> > > - Llama-3.1-8B
> > >
> > > |          | helpful-harmless                       | humour-harmless                        |
> > > |----------|----------------------------------------|----------------------------------------|
> > > | MOPO-FT   | $[0.31 \pm 0.01 \\, ; 0.34 \pm 0.02]$ | $[0.34 \pm 0.01 \\, ; 0.41 \pm 0.02]$ |
> > > | MOPO-LB  | $[0.45 \pm 0.02 \\, ; 0.43 \pm 0.05]$ | $[0.42 \pm 0.05 \\, ; 0.50 \pm 0.02]$ |
> > > | MOPO-Lag | $[0.48 \pm 0.02 \\, ; 0.46 \pm 0.03]$ | $[0.45 \pm 0.03 \\, ; 0.53 \pm 0.05]$ |
> > >
> > > - Qwen3-8B
> > >
> > > |          | helpful-harmless                       | humour-harmless                        |
> > > |----------|----------------------------------------|----------------------------------------|
> > > | MOPO-FT   | $[0.35 \pm 0.03  \\,; 0.38 \pm 0.01]$ | $[0.34 \pm 0.02 \\, ; 0.39 \pm 0.02]$ |
> > > | MOPO-LB  | $[0.48 \pm 0.02 \\, ; 0.51 \pm 0.04]$ | $[0.55 \pm 0.03 \\, ; 0.52 \pm 0.02]$ |
> > > | MOPO-Lag | $[0.52 \pm 0.03 \\, ; 0.53 \pm 0.02]$ | $[0.54 \pm 0.03 \\, ; 0.54 \pm 0.02]$ |
> > >
> > >
> > > As we can see, fixed temperatures result in a strictly weaker algorithm, which is consistent with the literature we cited in the rebuttal above. We will include full results in the revised version. We hope that we were able to fully clarify your concerns and invite you to kindly reconsider your score. Thanks a lot for the constructive discussion!

---

### Official Review · Reviewer_iS7v · 2026-03-13

**Soundness:** 3
**Presentation:** 2
**Significance:** 2
**Originality:** 2
**Overall Recommendation:** 4
**Confidence:** 3

**Summary:**

This paper introduces Multi-Objective Preference Optimization (MOPO), a novel algorithm designed for multi-objective preference learning. By solving the dual form of the constrained optimization problem inherent in this domain, the proposed approach converges to the Pareto front. This is achieved by first learning a data-dependent importance weighting function, which is subsequently used for negative log-likelihood (NLL) maximization.

**Compliance With Llm Reviewing Policy:**

Affirmed.

**Final Justification:**

The rebuttal satisfactorily addressed my primary concerns on lacking ablation studies and baselines.

**Key Questions For Authors:**

1. Reference Policy Lag: For the lagged reference policy, what motivated the choice of a constant lag instead of the more standard exponential moving average (EMA) of the online model? Could you discuss or provide empirical results on the impact of using an EMA update strategy with varying decay rates?
2. Modern Architecture Results: To further strengthen the paper's empirical claims, the authors might consider including results from more recent architectures (such as Qwen 3). Additionally, incorporating comparisons against stronger, contemporary multi-objective baselines would greatly enhance the evaluation and showcase the method's full potential.
3. Notation Corrections: There appears to be a few notation overlaps that slightly disrupt the readability of the mathematical derivations. For instance, $\chi$ is used to denote context in Line 123, but the same symbol seems to represent a different variable in Line 257.

**Limitations:**

Yes

**Strengths And Weaknesses:**

Strengths
1. Clarity and Presentation: The paper is generally well-written, with a logical flow and clear explanations of the underlying optimization problem.

2. Methodological Novelty: The approach of deriving and utilizing an optimal importance weighting function to balance competing objectives is an innovative contribution to preference alignment.

3. Theoretical Rigor: The proposed method is theoretically well-founded, providing a mathematically sound pathway to recovering Pareto-optimal policies.

Weaknesses:
1. Outdated Empirical Baselines: The main experiments rely on relatively small or older model architectures. The empirical claims would be significantly stronger if validated on modern, widely adopted models (e.g., Qwen 3 or DeepSeek R1).

2. Limited Scope in Evaluation: While the paper targets multi-objective preference learning, the primary experimental results only demonstrate the method's efficacy with a single secondary objective. This limits the ability to assess how the algorithm scales to more complex, real-world multi-objective scenarios (e.g., simultaneously balancing helpfulness, harmlessness, hallucinations, and humor)

3. Missing Contemporary Comparisons: The evaluation lacks comparisons with recent advancements in the multi-objective alignment space. To better contextualize the contribution, the authors should compare MOPO against contemporary frameworks such as Adaptive Multi-objective Preference Optimization (AMoPO)[1].

4. Insufficient Ablations: There is a lack of ablation studies examining the impact of the hyperparameter $\epsilon$.

[1] AMoPO: Adaptive multi-objective preference optimization without reward models and reference models

---

> ### Author Rebuttal · Authors · 2026-03-31
>
> Thank you very much for taking the time to review our paper. We try to address your concerns below:
>
> - **Weaknesses:**
> 1. Please note that while our contribution is more methodological and fundamental, whose scalability can be seen in our experiments on the 8B family of models, we understand your concern. Hence, we conducted additional experiments on Qwen3-8B [1]. Full results will be included in the revised version.
>
>     |                            | helpful-harmless                     | humour-harmless                      |
>     |----------------------------|--------------------------------------|--------------------------------------|
>     | RiC                        | $$[0.46 \pm 0.02 \\,\; 0.43 \pm 0.03]$$ | $$[0.50 \pm 0.01 \\,\; 0.48 \pm 0.04]$$ |
>     | PARM                       | $$[0.45 \pm 0.02 \\, \; 0.52 \pm 0.02]$$ | $$[0.52 \pm 0.02 \\,\; 0.50 \pm 0.01]$$ |
>     | MODPO                      | $$[0.38 \pm 0.01 \\, \; 0.40 \pm 0.03]$$ | $$[0.40 \pm 0.03 \\,\; 0.39 \pm 0.02]$$ |
>     | DPO | $$[0.42 \pm 0.03 \\,\; 0.38 \pm 0.02]$$ | $$[0.43 \pm 0.05 \\, \; 0.44 \pm 0.03]$$ |
>     | MOPO-LB                    | $$[0.48 \pm 0.02 \\, \; 0.51 \pm 0.04]$$ | $$[0.55 \pm 0.03  \\, \; 0.52 \pm 0.02]$$ |
>     | MOPO-Lag                   | $$[0.52 \pm 0.03  \\, \; 0.53 \pm 0.02]$$ | $$[0.54 \pm 0.03 \\, \; 0.54 \pm 0.02]$$ |
>     | AMoPO                      | $$[0.44 \pm 0.04 \\, \; 0.50 \pm 0.05]$$ | $$[0.50 \pm 0.05 \\, \; 0.50 \pm 0.03]$$ |
>
>
> 2. Please note that to show MOPO’s scalability to more than two objectives, we have also conducted three objective experiments (Table 2 and Table 6 show results on helpful-humour-harmless objectives). Even in the case of three objectives, we observe that MOPO consistently performs as good as or better than baselines. This shows that the number of objectives is not a limitation of the MOPO algorithm.
> 3. Thank you for providing a reference to AMoPO. We were unaware of this work. We have conducted some initial experiments. Please see the table above. We can see that while competitive, AMoPO is outperformed by PARM and MOPO on at least two pairs of objectives.
> 4. Please see our response to Reviewer SmiK (Question 1).
>
>
> - **Questions:**
> 1. Thank you for pointing this out. It indeed is an interesting ablation study to be done. We conducted initial experiments and present the results below. We compare constant lag reference policy updates on Qwen3-8B with different EMA decay factors ($\zeta$):
>
>     |                              | helpful-harmless                     | humour-harmless                      |
>     |------------------------------|--------------------------------------|--------------------------------------|
>     | $$\zeta = 0.9999$$           | $$[0.49 \pm 0.01 \\,\; 0.50 \pm 0.01]$$ | $$[0.51 \pm 0.01 \\,\; 0.49 \pm 0.02]$$ |
>     | $$\zeta = 0.99$$             | $$[0.51 \pm 0.02 \\,\; 0.53 \pm 0.03]$$ | $$[0.53 \pm 0.03 \\,\; 0.54 \pm 0.02]$$ |
>     | $$\zeta = 0.9$$              | $$[0.48 \pm 0.06 \\,\; 0.49 \pm 0.05]$$ | $$[0.51 \pm 0.05 \\,\; 0.49 \pm 0.05]$$ |
>     | Constant lag $$t_{0} = 500$$ | $$[0.52 \pm 0.03 \\,\; 0.53 \pm 0.01]$$ | $$[0.53 \pm 0.03 \\, \; 0.54 \pm 0.02]$$ |
>
>
> 2. Please see above
> 3. Please note that in Line 123 we are talking about the context space $\mathcal{X}$ (\mathcal{X}), whereas in Line 257 we are talking about the dual variable $\chi$ (\chi). While they might look similar, they represent different quantities. We will look into using more visually distinct symbols for the reader. Thank you for pointing this out.
>
> &nbsp;
>
> Please let us know if you have any more questions, and we would be happy to discuss. We hope that our clarifications have satisfactorily addressed your comments, and we kindly invite you to reconsider your assessment. We sincerely appreciate your time and feedback.
>
> &nbsp;
>
> [1] Yang, An, et al. "Qwen3 technical report." arXiv preprint arXiv:2505.09388 (2025).

---

> > ### Author Rebuttal · Reviewer_iS7v · 2026-04-05
> >
> > I appreciate the authors' efforts during the rebuttal. Most of my concerns have been effectively resolved, and I have updated my score accordingly

---

### Official Review · Reviewer_AfB9 · 2026-03-14

**Soundness:** 3
**Presentation:** 2
**Significance:** 4
**Originality:** 4
**Overall Recommendation:** 5
**Confidence:** 2

**Summary:**

This paper proposes Multi-Objective Preference Optimization: a regularized concave optimization problem is solved with regard to one objective, while all other objective(s) are constrained with a lower bound. The authors claimed that the proposed method can achieve better approximation of the true Pareto frontier than baseline methods such as DPO on synthetic and LLM-related tasks.

**Compliance With Llm Reviewing Policy:**

Affirmed.

**Final Justification:**

I recommend the acceptance of this paper.

**Key Questions For Authors:**

1. Can you show scalarization as a baseline for the metrics reported in Table 2?
2. How would you recommend practitioners to set proper constraint values? What happens when there is no "natural" constraint value? Would you recommend them to err on the higher or the lower end? How sensitive is your recommendation over different sets of objectives and true Pareto fronts?

**Limitations:**

Yes, the authors adequately discussed the limitations and potential negative societal impact of their work.

**Strengths And Weaknesses:**

Strengths:
1. Soundness: This paper includes extensive derivation and theoretical analysis of the proposed optimization formulation.
2. Soundness: The authors conducted synthetic experiments where the ground truth Pareto fronts are known.
3. Significance: This paper addresses the important challenges in the alignment of LLMs when multiple objectives are present without natural scalarization.
4. Originality: This paper finds a novel application of concave constrained optimization formulations for multi-objective optimization.

Weaknesses:
1. Soundness: The authors did not include scalarization as a baseline method.
2. Soundness: The proposed method did not outperform the PARM baseline by a significant margin on the majority of the reported tasks.
3. Presentation: This paper is very dense, sometimes with acronyms and concepts used without proper introduction (e.g., "COP" was used on page 2 and later introduced in page 3). The scope of this work is larger than the space constraint of a conference paper. It might help readers if only the core ideas are presented in the main text.

---

> ### Author Rebuttal · Authors · 2026-03-31
>
> We wish to thank you for your time in reviewing our paper and  try to address your concerns below:
>
> - **Weaknesses:**
> 1. Please note that in current literature, scalarization is one of the weakest baselines and has been outperformed by methods like PARM, RiC, MORLHF, and MODPO [1], just to name a few. We already include many of these superior methods in our experiments, hence comparison with scalarization methods would be redundant.
>
> 2 and 3. Thank you for these comments, we will try to incorporate them in the revised version.
>
> - **Questions:**
> 1. Please see above
> 2. We view constraint selection in MOPO as choosing a target point on the Pareto frontier rather than fixing absolute thresholds. In practice, for each objective $k$, we set $b_k$ relative to a reference policy (e.g., SFT or RM-optimized baseline) as $b\_k \approx \mathbb{E}\_{y \sim \pi\_{\mathrm{ref}}, y' \sim \mu}[q\_{k}(y \succ y')] - \delta\_k$, where $\delta\_k \ge 0$ encodes the acceptable degradation. When no natural constraint exists, we recommend sweeping $b_k$ (or equivalently, the dual variables) to trace the empirical Pareto frontier and selecting operating points via downstream validation metrics, which avoids committing to arbitrary thresholds (this is similar to what is done in empirical constrained optimization methods). In terms of misspecification, we advise erring on the stricter (higher) constraint side: if $b_k$ is too low, the constraint becomes inactive ($\lambda_k \to 0$) and MOPO effectively reduces to single-objective optimization, whereas if $b_k$ is higher, the method remains stable and yields a conservative solution. This induces an asymmetric sensitivity where overly loose constraints are more harmful than overly strict ones. Finally, sensitivity across objectives depends on the curvature of the Pareto front, where performance is usually robust in flat regions. However, MOPO’s primal–dual updates adaptively reweight objectives via $\lambda_k$, providing local stability to moderate perturbations in $b_k$. Empirically, a coarse grid over $b_k$ suffices to recover the relevant Pareto region across diverse objectives, as also seen in Figure 6.
>
> &nbsp;
>
> Please let us know if you have any more questions, we would be happy to discuss. Thank you
>
> &nbsp;
>
> [1] Z. Zhou, et al. "Beyond one-preference-fits-all alignment: Multi-objective direct preference optimization." ACL 2024.

---

> > ### Author Rebuttal · Reviewer_AfB9 · 2026-04-02
> >
> > Thank you for your rebuttal. I don’t have any further questions. Please address 2 and 3 in the camera-ready version if possible.

---

### Official Review · Reviewer_SmiK · 2026-03-16

**Soundness:** 3
**Presentation:** 3
**Significance:** 3
**Originality:** 3
**Overall Recommendation:** 5
**Confidence:** 4

**Summary:**

This paper proposes MOPO, an offline method for aligning LLMs with multiple preference objectives. The idea is to cast multi-objective alignment as a concave constrained optimization problem: maximize a primary preference objective subject to lower-bound constraints on secondary objectives, with KL regularization to a reference policy. MOPO works directly on pairwise preference data. On the theory side, they prove that merging/decoding approaches using non-barrier f-divergences cannot achieve full Pareto optimality, which motivates the barrier-function-based formulation. The experiments cover synthetic data and text generation tasks.

**Compliance With Llm Reviewing Policy:**

Affirmed.

**Final Justification:**

The rebuttal addresses my main concerns and reinforces my prior positive assessment. I am maintaining my recommendation of 4, with increased confidence in the soundness of the method. I encourage the authors to incorporate the convergence theorem, the ablation table, and the Panacea/L3Ms discussion into the camera-ready version as promised. Thank you again for the careful and substantive response.

**Key Questions For Authors:**

1. **Ablation studies. ** Can you do some ablation studies over the KL-ball radius \epsilon? This parameter controls the conservativeness of the lower-bound estimation and I want to know how sensitive performance is to this choice, and whether the estimated lower bounds are actually calibrated.

2. Could you address the questions on the missing baselines or clarify why they may be out of scope in this work?

**Limitations:**

Yes.

**Strengths And Weaknesses:**

Strengths:

1, **The constrained optimization formulation is well-motivated.** I think the framing of multi-objective alignment as a concave constrainted optimization problem makes sense, and the theories (A.2.4, A.2.6) are very solid.  The connection between log-barrier functions and Pareto optimality (Remark 3.2, page 4) is the kind of observation I wish more alignment papers would make as it clarifies why the formulation should work, not just that it does.

2, **Experiments are comprehensive.** Five different 7B-class LLMs, two real-world tasks with four reward-pair configurations over 5 runs. This is more comprehensive than what I usually see in this type of work.

3, **Fully offline, no reward model needed. **

Weakness:

1, **Some gaps between the theory and what's actually implemented.** The Pareto optimality guarantee needs you to know the Pareto front to set constraint thresholds which I think is circular.  The alternating optimization in Algorithm 1 also lacks convergence analysis as the log-barrier variant's convergence "is not guaranteed" for their setting (page 21, line 1157), and the Lagrangian variant gets no formal analysis at all.

2, **Missing baselines.** Panacea [1] and L3Ms [2] are both directly relevant and were published before the submission deadline. Panacea claims to recover the entire Pareto front via SVD-based low-rank adaptation but it was never compared against in this work and L3Ms uses log-barrier Lagrangian optimization for LLM alignment constraints, which makes it the obvious comparison for MOPO-LB.

Reference:

[1] Panacea: Pareto Alignment via Preference Adaptation for LLMs
[2] L3Ms: Lagrange Large Language Models

---

> ### Author Rebuttal · Authors · 2026-03-31
>
> We wish to thank you for your time in carefully reviewing our paper. We try to address your questions below:
>
> - **Weaknesses**:
> 1. While we were unable to include theoretical results at the time of submission due to time constraints, we briefly present a proof sketch below, full proof of which will be included in the revised version.
>
>     **Theorem.** Consider the saddle-point optimization of the Lagrangian $L(\rho, \lambda)$ in Line 168. Let $(\rho^\star, \lambda^\star)$ be the unique saddle point. Under optimistic primal-dual dynamics with step sizes $\eta_\rho, \eta_\lambda > 0$, the iterates as defined in Algorithm 1 converge with some $c \in (0,1)$ as:
>
>     $$\Phi_t := \text{KL}(\rho^\star || \rho_t) + \frac{1}{2\eta_\lambda} ||\lambda^\star - \lambda_t ||_2^2 \le \Phi_0 (1 - c)^{t}. $$
>
>     **Proof sketch**.  The negative entropy $-\tau H(\rho)$ in the MOPO objective renders $L(\rho, \lambda)$ $\tau$-strongly concave in $\rho$ with respect to the KL divergence. This implies that for any $\lambda$:
>
>     $$L(\rho^\star, \lambda) - L(\rho, \lambda) \ge \langle \nabla_\rho L(\rho, \lambda), \rho^\star - \rho \rangle + \tau \text{KL}(\rho^\star || \rho)$$
>
>     This variational geometry ensures that the primal space is "curved" toward $\rho^\star$, allowing for faster-than-sublinear rates. Defining $\Phi_t$ as the combined primal-dual distance to the optimum, and applying Mirror-Prox analysis [1], we evaluate the progress per step:
>
>     $$\Phi_{t} \le \Phi_{t-1} - \eta_\rho \left( L(\rho_t, \lambda^\star) - L(\rho^\star, \lambda_t) \right) + \text{ErrorTerms}$$
>
>     By choosing $\eta_\rho < 1/L_{smooth}$ with $L_{smooth}$ being the Lipschitz gradient constant, we can ignore the error terms. The $\tau$-strong concavity allows us to bound the duality gap by the current distance: $L(\rho_t, \lambda^\star) - L(\rho^\star, \lambda_t) \ge c' \Phi_t$. This yields the linear contraction: $\Phi_t \le (1 - c) \Phi_{t-1} \implies \Phi_t \le \Phi_0 (1-c)^t$. This result: (i) guarantees the final importance ratio (policy) $\rho_T$ is optimal, (ii) confirms why MOPO does not oscillate around constraints in high-dimensional settings, and (iii) shows how with learned policies, convergence holds up to an error of $O(\varepsilon_{\text{approx}} + \varepsilon_{\text{stat}})$.
>
> 2. Thank you for pointing out these references. Please note that PARM and RiC outperform Panacea on a variety of tasks [2,3]. Regarding L3Ms, we were unaware of this work so thank you for pointing this out. Unfortunately, we could not find an open-source implementation of this method (the original repository is empty:  https://github.com/Guneet-Dhillon/l3m), and within this limited rebuttal period, we are unable to conduct experiments with L3Ms.
>
> - **Questions**:
> 1. We conducted some initial ablation experiments on Llama-3.1-8B in the limited time we had. Full results will be included in the revised version. Overall, we observe that with smaller $\epsilon$ we have tighter, more accurate lower bounds, implying stronger constraint enforcement, whereas with larger $\epsilon$, we get overly conservative policies in which constraints become vacuous, and performance collapses toward single-objective optimization.
>
>      |                    | helpful-harmless                     | humour-harmless                      | no hallucinate-faithful              |
>     |--------------------|--------------------------------------|--------------------------------------|--------------------------------------|
>     | $$\epsilon=0.005$$ | $$[0.42 \pm 0.02 \\, \; 0.47 \pm 0.01]$$ | $$[0.46\pm 0.03\\,\; 0.45 \pm 0.02]$$  | $$[0.43 \pm 0.02\\,\; 0.51 \pm 0.01]$$ |
>     | $$\epsilon=0.01$$  | $$[0.45 \pm 0.02 \\, \; 0.42 \pm 0.02]$$ | $$[0.47 \pm 0.02 \\, \; 0.49 \pm 0.04]$$ | $$[0.47 \pm 0.03 \\,\; 0.49 \pm 0.02]$$ |
>     | $$\epsilon=0.05$$  | $$[0.38 \pm 0.01 \\, \; 0.33 \pm 0.03]$$ | $$[0.44 \pm 0.03 \\, \; 0.40 \pm 0.01]$$ | $$[0.45 \pm 0.03 \\, \; 0.40 \pm 0.03]$$ |
>     | $$\epsilon=1$$     | $$[0.42 \pm 0.05 \\, \; 0.28 \pm 0.06]$$ | $$[0.43 \pm 0.04 \\, \; 0.35 \pm 0.06]$$ | $$[0.43 \pm 0.05 \\, \; 0.39 \pm 0.04]$$ |
>
> 2. Please see above
>
> &nbsp;
>
> Please let us know if you have any more questions, and we would be happy to discuss. We hope that we were able to address your concerns. Thank you
>
> &nbsp;
>
> [1] A. Nemirovski. "Prox-method with rate of convergence O (1/t) for variational inequalities with Lipschitz continuous monotone operators and smooth convex-concave saddle point problems." SIAM Journal on Optimization, 2004.
>
> [2] PAD: Personalized Alignment of LLMs at Decoding-time. ICLR 2025.
>
> [3] PARM: Multi-Objective Test-Time Alignment via Preference-Aware Autoregressive Reward Model. ICML 2025.

---

> > ### Author Rebuttal · Reviewer_SmiK · 2026-04-04
> >
> > Thank you for addressing my concerns! I will maintain my positive assessment of this submission.

---

### Decision · Program_Chairs · 2026-04-30

**Decision:**

Accept (regular)

**Comment:**

This paper proposes an offline method for multi-objective LLM alignment by formulating preference optimization as a concave constrained problem, where a primary objective is maximized subject to some constraints, together with KL regularization to a reference policy. Most reviewers found this formulation well motivated, appreciated its connection to Pareto optimality, and viewed the empirical evaluation as reasonably comprehensive across both synthetic settings and real text-generation tasks.

The main reservations concerned the gap between the theory and the practical optimization procedure, as well as missing or initially underdeveloped baselines and ablations. Presentation was also a recurring issue: several reviewers felt that the paper is dense, not fully polished, and would benefit from a clearer exposition of the core ideas and more concrete practical guidance on how to set the constraints.

Overall, the reviewer scores are roughly positive, though not unanimously enthusiastic. One reviewer remained less convinced even after the rebuttal and continued to raise concerns during the discussion phase. The rebuttal appears to have addressed several technical issues through additional experiments and clarifications, many of which were promised for the revised version, although naturally their final integration cannot be fully verified at this stage.